# Taxonomic and Phylogenetic Characterizations Reveal Four New Species, Two New Asexual Morph Reports, and Six New Country Records of Bambusicolous *Roussoella* from China

**DOI:** 10.3390/jof8050532

**Published:** 2022-05-20

**Authors:** Dong-Qin Dai, Nalin N. Wijayawardene, Monika C. Dayarathne, Jaturong Kumla, Li-Su Han, Gui-Qing Zhang, Xian Zhang, Ting-Ting Zhang, Huan-Huan Chen

**Affiliations:** 1Center for Yunnan Plateau Biological Resources Protection and Utilization, Yunnan Engineering Research Center of Fruit Wine, College of Biological Resource and Food Engineering, Qujing Normal University, Qujing 655011, China; cicidaidongqin@gmail.com (D.-Q.D.); nalinwijayawardene@yahoo.com (N.N.W.); hanlisurain@outlook.com (L.-S.H.); zhangguiqingaoligei@outlook.com (G.-Q.Z.); fungixianzhang@outlook.com (X.Z.); torizhangtingting@outlook.com (T.-T.Z.); 2Section of Genetics, Institute for Research and Development in Health and Social Care, No: 393/3, Lily Avenue, Off Robert Gunawardane Mawatha, Battaramulla 10120, Sri Lanka; 3Postgraduate Institute of Agriculture (PGIA), University of Peradeniya, Peradeniya 20400, Sri Lanka; monidaya40@gmail.com; 4Research Centre of Microbial Diversity and Sustainable Utilization, Chiang Mai University, Chiang Mai 50200, Thailand; jaturong_yai@hotmail.com

**Keywords:** bambusicolous ascomycetes, new records, new taxa, phylogeny, taxonomy

## Abstract

During the ongoing investigation of bambusicolous ascomycetous fungi in Yunnan, China, 24 specimens belonging to the family *Roussoellaceae* were collected and identified based on morphological features and phylogenetic support. Maximum-likelihood (ML) analyses and Bayesian analyses were generated based on the combined data set of ITS, LSU, *tef1*, and *rpb2* loci. The phylogenetic analyses revealed four novel lineages in *Roussoella s. str.*; thus, we introduced four new species *viz.*, *Roussoella multiloculate* sp. nov., *R. papillate* sp. nov., *R. sinensis* sp. nov., and *R*. *uniloculata* sp. nov. Their morphological characters were compared with the known *Roussoella* taxa, which lack sequence data in the GenBank. Asexual morphs of *R. kunmingensis* and *R. padinae* were recorded from dead bamboo culms in China (from the natural substrates) for the first time. *Neoroussoella bambusae*, *Roussoella japanensis*, *R. nitidula*, *R. padinae*, *R. scabrispora,* and *R. tuberculate* were also reported as the first records from China. All new taxa are described and illustrated in detail. Plates are provided for new reports.

## 1. Introduction

The family *Roussoellaceae* Liu et al. [1] accommodates three genera, *viz.*, *Neoroussoella* Liu et al., *Roussoella* Sacc. and *Roussoellopsis* Hino and Katum. Later, Jaklitsch and Voglmayr [2] synonymized *Roussoellaceae* under *Thyridariaceae* based on the multigene analysis of limited taxa. Tibpromma et al. [3] reinstated *Roussoellaceae*; treating *Roussoellaceae* and *Thyridariaceae* as distinct families within *Pleosporales*. Subsequent studies confirmed this separation using additional taxa and combining morphological and phylogenetic analyses [4,5,6,7,8]. Currently, twelve genera have been accepted (*viz.*, *Appendispora* Hyde, *Cytoplea* Bizz. and Sacc., *Elongatopedicellata* Zhang et al., *Immotthia* Barr, *Neoroussoella*, *Pararoussoella* Wanas. et al., *Pseudoneoconiothyrium* Wanas et al., *Pseudoroussoella* Mapook and Hyde, *Roussoella* Sacc., *Roussoellopsis* Hino and Katum., *Setoarthopyrenia* Mapook and Hyde and *Xenoroussoella* Mapook and Hyde) in *Roussoellaceae* [9].

*Roussoella* (the type of genus of *Roussoellaceae*) was introduced by Saccardo and Paoletti [10] with *R. nitidula* Sacc. and Paol. as the type species recorded from bamboo in Malacca, Malaysia. Höhnel [11] proposed that *Dothidea hysterioides* Ces. is the former name for this taxon; thus, being transferred to the *Roussoella* genus, i.e., *Roussoella hysterioides* (Ces.) Höhn. However, Aptroot [12] and Müller and Arx [13] assigned *Roussoella* to *Amphisphaeriaceae*, which comprises cylindrical and unitunicate asci, immersed ascostromata, two-celled and brown ascospores. Aptroot [12] described *Roussoella* asci as unitunicate and moved the three species to this genus, while Aptroot [14] modified his concept of *Roussoella* asci and considered them as bitunicate. *Roussoella* is characterized by immersed, large, and loculate ascostromata, cylindrical and bitunicate asci, brown, fusiform to ellipsoidal, ornamented, and two-celled ascospores surrounded by a sheath [1,15,16]. *Roussoella* mainly occurs on gramineous plants, with most species found on bamboo and palms [1,16,17].

*Neoroussoella* is a monotypic genus, distinct from *Roussoella*, with uniloculate ascomata, absence of a clypeus, and an asexual morph forming hyaline to brown, oblong to ellipsoidal, and smooth-walled phoma-like conidia [1,18]. Further, *Neoroussoella* is characterized by a distinct asexual morph producing relatively smaller (3–4 × 1.5–2 μm), hyaline conidia with smooth wall [1]. Currently, this genus comprises eleven species [19,20].

The aims of this study are to introduce four new species, *viz*. *R. multiloculate*, *R. papillate*, *R. sinensis*, and *R. uniloculata*. In addition, the asexual morphs of *R. kunmingensis* and *R. padinae* isolated from dead bamboo culms (from a natural substrate) are also described for the first time. Finally, the species *Neoroussoella bambusae*, *Roussoella japanensis*, *R. nitidula*, *R. padinae*, *R. scabrispora,* and *R. tuberculate* are reported for the first time in China.

## 2. Materials and Methods

### 2.1. Fungal Sampling and Morphology

Bamboo culms were collected in Yunnan, China, and conserved in protection bags for two days until they arrived at the laboratory. Samples were examined, and single spore isolation was performed as previously described [17]. Morphological characters were examined using water slides and photographed (Olympus BX53 DIC compound microscope with an Olympus DP74 camera). Fruiting bodies, such as ascostromata and conidiomata, were also photographed (Leica S8AP0 stereomicroscope with an HDMI 200C camera). Measurements were registered (Tarosoft (R) Image Frame Work 80 software). Specimens and living cultures were deposited at the Herbarium of Guizhou Medical University (GMB) and Guizhou Medical University Culture Collection (GMBCC) in Guiyang, China. Duplicates of holotypes and ex-type cultures were also deposited at the herbarium of Research Institute of Resource Insects, Chinese Academy of Forestry (IFRD), and Research Institute of Resource Insects, Chinese Academy of Forestry Culture Collection (IFRDCC) in Kunming, China. Index Fungorum [20] numbers were provided for newly introduced taxa.

### 2.2. DNA Extraction, Polymerase Chain Reaction (PCR) Amplification, and Phylogeny

Pure cultures were grown on PDA media, for 30–40 days, at 28 °C, in the dark. Fresh mycelium was scraped using a surgical knife and placed into a 1.5 mL centrifuge tube, and ground into powder using liquid nitrogen. Genomic DNA was extracted following the instruction book of the Biospin Fungus Genomic DNA Extraction Kit (BioFlux^®^).

Information of primers used for the amplification of internal transcribed spacers (ITS), small subunit rDNA (SSU), large subunit rDNA (LSU), translation elongation factor 1-α gene region (*tef1*), and RNA polymerase II second largest subunit (*rpb2*) genes is presented in Table 1. ITS, SSU, LSU, *tef1,* and *rpb2* loci amplifications were performed by polymerase chain reaction (Eppendorf Mastercycler nexus gradient) according to the conditions presented in Table 2 [17]. Both forward and reverse primers (Table 1) were used for sequencing and these primers were the same as those used for amplification. The PCR products were sequenced, and the sequences were deposited in GenBank, as shown in Table 3.

### 2.3. Phylogenetic Analyses

The quality of the sequences was verified (BioEdit v.7.0 [25]), and alignments from single genes were generated (MAFFT v.7.215 [26]) (http://mafft.cbrc.jp/alignment/server/index.html, accessed on 20 April 2021), being manually edited when needed (MEGA6 version 6.0 [27,28]). The combined alignment of multi-genes was carried out (MEGA6 [27]). Maximum-likelihood (ML) analyses were performed (software RAxMLGUI v.1.0 [29,30]) with a 1000 bootstrap. Multi-gene alignments were uploaded to the website (http://sing.ei.uvigo.es/ALTER/, accessed on 20 April 2022) to obtain the PHYLIP format file. The best nucleotide substitution model was determined using the online tool Findmodel (http://www.hiv.lanl.gov/content/sequence/findmodel/findmodel.html, accessed on 20 April 2022) and was executed in RAxMLGUI to generate the best ML.

Bayesian analyses were performed using MrBayes v.3.0b4 [31]. MrModeltest v.2.2 selected the best evolution model [32]. Posterior probabilities (PP) [33,34] were performed by Markov Chain Monte Carlo sampling (MCMC) [35]. Six simultaneous Markov chains were run for 1,000,000 generations, and trees were sampled every 100th generation. The burn-in was set to 0.25, and the run was automatically stopped when the average standard deviation of split frequencies reached below 0.01 [36].

Trees were constructed (TreeView [37]) and formatted (Adobe Illustrator CS v.5). Maximum-likelihood bootstrap values (MLBP) equal to or greater than 50% and Bayesian posterior probabilities (BYPP) > 0.80 are given at the branches. The sequences used in this study are listed in Table 1. The alignment based on the combined loci and phylogenetic tree was submitted to TreeBASE under the code 29601 (https://www.treebase.org/, accessed on 20 April 2022).

## 3. Results

### 3.1. Phylogenetic Analyses

The sequence data set of combined ITS, LSU, *tef1,* and *rpb2* loci was used to determine the phylogenetic position of the newly generated described strains. SSU sequences were not included in the alignment, as most *Roussoella* taxa lack SSU in the GenBank. The dataset comprised 90 strains, including two outgroup strains (*Torula herbarum* CBS 220.69 and CBS 111855, Table 3). The final alignment comprises 3381 characters used for the phylogenetic analyses, including gaps. The RAxML analysis of the combined dataset generated a best-scoring tree with a final ML optimization likelihood value of −30,685.326261 (Table 4). GTR+I+G model was selected as the best model based on MrModeltest and was used for the Bayesian analysis.

Based on the multi-gene phylogenetic analyses (Figure 1), 26 new isolates were grouped in the family *Roussoellaceae* (96% MLBP, 1.00 BYPP). Five isolates, GMBCC1056, GMBCC1065, GMBCC1069, GMBCC1071, and GMBCC1080, and one specimen, GMB1219, represented a novel species *Roussoella multiloculate* sp. nov. and formed a sister clade to *R. verrucispora* with high statistical support (100% MLBP, 1.00 BYPP). *Roussoella papillate* sp. nov. (GMBCC 1121 and IFRDCC 3103) grouped sister with *R. japanensis* and *R. hysterioides* with high bootstrap support (100% MLBP, 1.00 BYPP). The third new species, *R. sinensis* sp. nov., clustered together with *R. doimaesalongensis*, *R. siamensis,* and *R. yunnanensis*. The last new taxon, *R. uniloculata*, forms a distinct clade at the base of lineage, which contains *R. angusta*, *R. chiangraina*, *R. kunmingensis*, *R. magnatum*, *R. mediterranea*, *R. neopustulans,* and *R. padinae*. The *R. sinensis* and *R. uniloculata* clades are phylogenetically distant from the known species (Figure 1).

### 3.2. Taxonomy

***Neoroussoella bambusae*** Phook., Liu and Hyde, Phytotaxa 181(1):23 (2014) Figure 2.

Index Fungorum number: IF 550669.

For descriptions of sexual and asexual morphs see Liu et al. [1].

Distributions: Thailand, China.

Material examined: China, Yunnan Province, Mengla, Jinghong, Xishuangbanna Topic Botany Garden, Bamboo Garden, (101°24′41.74″ N, 21°93′40.94″ E, 507.88 m), on dead culms of bamboo, 15 August 2020, Dong-Qin Dai, DDQ01025 (GMB1291); living culture, GMBCC1116, GenBank accession number SSU: OM891830; *Ibid*. DDQ01044 (GMB1295); living culture, GMBCC1118 (new country record), GenBank accession number SSU: OM891832.

Notes: In morphology, GMB1291 and GMB1295 are similar to each other and resemble *N. bambusae*. Furthermore, LSU, ITS, and *rpb2* gene regions of GMBCC1116 and GMBCC1118 are identical to each other. Therefore, new collections GMBCC1116 and GMBCC1118 are identified as *N. bambusae*. Our new strains (GMBCC1116 and GMBCC1118) grouped in a clade comprising *N. alishanense* (FU31016, ex-type), *N. bambusae* (MFLUCC 11-0124, ex-type) and *R. arundinacea* (CBS 146088) with high statistical support (100% MLBP, 1.00 BYPP) (Figure 1). *Neoroussoella bambusae* differs from *N. alishanense* in having ascostromata that are visible as black dome-shaped or shield-shaped or flattened ovoid areas on the host surface while *N. alishanense* is *saprobic* on *Pennisetum purpureum* and having hemispherical to subconical ascomata and ascospores which lack mucilaginous sheath around the ascospores [18]. This is the first report of this species from China.

***Roussoella japanensis*** Kaz. Tanaka, Liu and Hyde, Phytotaxa 181(1):14 (2014) Figure 3.

Index Fungorum number: IF 550663.

For descriptions of sexual and asexual morphs see Liu et al. [1].

Distributions: Japan, China.

Material examined: China, Yunnan Province, Kunming Expo Park, on dead culms of bamboo, 24 July 2020, Dong-Qin Dai, DDQ00780 (GMB1220), living culture, GMBCC1067 (new country record), GenBank accession number SSU: OM891823; Luoping, Jiulong waterfall, on dead culms of bamboo, 29 August 2020, Dong-Qin Dai, DDQ01029 (GMB1292), living culture, GMBCC1117, GenBank accession number SSU: OM891831.

Notes: Our phylogenetic analyses showed that new strains GMBCC1067 and GMBCC1117 clustered together with *R. japanensis* (MAFF 239636, ex-type) with strong statistical supports (100% MLBS, 1.00 BYPP) (Figure 1). Furthermore, there are no base pair differences in ITS, *tef1* and *rpb2* loci of GMBCC1067, GMBCC1117, and MAFF 239636 which indicates that they are conspecific. Morphologically, they are identical to each other [1]. Hence, based on both morphology and phylogeny, we identified GMBCC1067 and GMBCC1117 as *R. japanensis*. *Roussoella japanensis* was introduced by Liu et al. [1] from twigs of *Sasa veitchii* var. *veitchii* in Japan. This is the first record of *R. japanensis* from China.

***Roussoella kunmingensis*** H.B. Jiang, Phookamsak and Hyde, Mycol. Prog. 18(4):581 (2019) Figure 4.

Index Fungorum number: IF 827562.

*Saprobic* on decaying bamboo culms. Sexual morph: see Jiang et al. (2019). Asexual morph: *Stromata* forming under a light brown area, up to 250–300 µm long and 75–105 µm wide, and becoming raised at maturity, globose, ellipsoidal to irregular. *Conidiomata* 55–75 µm high, 200–300 µm diam., locules solitary immersed in the stromata, globose to subglobose, dark brown. *Conidiomatal wall* comprising two layers of cells of *textura angularis*, with dark to brown outer thin layer 6–8 µm thick, with a 12–21 µm thick, hyaline, conidiogenous inner layer. *Conidiophores* reduced to conidiogenous cells. *Conidiogenous cells* 3.5–6.5 × 3–5 µm ($\bar{x}$ = 4.7 × 3.9 µm, n = 5) enteroblastic, phialidic, indeterminate, discrete, ampulliform, hyaline, smooth-walled. *Conidia* 4–5 × 2.5–3.5 µm ($\bar{x}$ = 4.3 × 2.9 µm, n = 20), ellipsoidal, oblong, aseptate, straight, rounded at both ends, hyaline when immature, and becoming brown to dark brown when mature, smooth-walled, inside usually containing 1–2 small guttules.

Culture characters: Ascospores and conidia germinating on PDA within 24 h and germ tubes produced from both cells and both sides. Colonies slow-growing, 15 mm diam. after 20 days at 28 °C, under 24 h dark, circular, floccose at the centre, with even margin, white at margin, and light yellow at the centre.

Distributions: China.

Material examined: China, Yunnan Province, Kunming, Kunming Expo Park (25°07′77″ N, 102°76′23″ E, 1960 m), on dead culms of bamboo, 24 July 2020, Dong-Qin Dai, DDQ00742, GMB1203, living culture, GMBCC1055; *Ibid*., DDQ00745 (GMB1206), living culture GMBCC1057, GenBank accession number SSU: OM891820; Diqin, Shangri-La, Bigu Mountain, on dead culms of bamboo, 22 July 2020, 27°36′56.9″ N, 99°42′6.4″ E, 3460 m, Dong-Qin Dai, DDQ00905, GMB1259; living culture, GMBCC1086 (first report of the asexual morph), GenBank accession number SSU: OM891825.

Notes: *Roussoella kunmingensis* is characterized by having immersed, uniloculate ascomata, cylindrical to cylindric-clavate, bitunicate asci, and ellipsoidal to fusiform, light brown to brown, 2-celled ascospores with longitudinal ribs. Multi-gene phylogenetic analyses showed that GMBCC1055, GMBCC1057, and GMBCC1086 grouped with *R. kunmingensis* (KUMCC 18-0128, ex-type) with high statistic values (100% MLBP, 1.00 BYPP). Base pair arrangement of ITS, *tef1*, and *rpb2* regions of strains GMBCC1055, GMBCC1057, and GMBCC1086 and KUMCC 18-0128 were identical. However, Jiang et al. [5] described this species with only sexual morph from bamboo in China. Here, one new collection with sexual morph and two new collections with asexual morph were examined. Hence, we reported the asexual morph of *R. kunmingensis* for the first time and provide a description of the asexual morph and an illustration of the holomorph.

***Roussoella multiloculate.*** Dai and Wijayaw., sp. nov. Figure 5.

Index Fungorum number: IF 556005.

Etymology: Reference to its multi-loculate conidiomata.

Holotype: GMB1207.

*Saprobic* on dead bamboo branches. Sexual morph: Undetermined. Asexual morph: *Stromata* forming under a blackened area, up to 0.7–1.5 mm long and 0.3–0.5 mm wide, and becoming raised at maturity, ellipsoidal to oblong, occasionally irregular. *Conidiomata* 30–110 µm wide, 50–1000 µm long, loculate, 3–10 locules gregarious immersed in the stromata, globose to subglobose, dark brown, with slit-like opening. *Conidiomatal wall* comprising several layers of cells of *textura angularis*, with dark to brown outer thin layer, 5–7 µm thick, with 10–15 µm thick, hyaline, conidiogenous inner layer. *Conidiophores* reduced to conidiogenous cells. *Conidiogenous cells* 2.5–10 × 2–5 µm ($\bar{x}$ = 8.7 × 3.1 µm, n = 20) enteroblastic, phialidic, indeterminate, discrete, cylindrical to ampulliform, hyaline, smooth-walled. *Conidia* 4–5.5 × 2.5–3.7 µm ($\bar{x}$ = 5.1 × 3.1 µm, n = 20), ellipsoidal oblong, aseptate, straight, rounded at both ends, hyaline when immature and becoming brown to dark brown when mature, smooth-walled, inside usually containing 2 small guttules.

Culture characters: Conidia germinating on PDA within 24 h and germ tubes produced from one side. Colonies slow growing, 40 mm diam. after 20 days at 28 °C, under 24 h dark, circular, with even margin, floccose at the center, greenish brown, and white at margin, ring-like from below.

Distributions: China.

Material examined: China, Yunnan Province, Kunming, Kunming Expo Park (25°07′77″ N, 102°76′23″ E, 1960 m), on dead culms of bamboo, 24 July 2020, Dong-Qin Dai, DDQ00748 (GMB1207, holotype), ex-type GMBCC1056, GenBank accession number SSU: OM891821; *Ibid*. (IFRD500-21 isotype), ex-isotype IFRDCC 3100; *Ibid*. DDQ00774 (GMB1218), living culture, GMBCC1065, GenBank accession number SSU: ON124715; *Ibid*. DDQ00775 (GMB1219), GenBank accession number SSU: OM891822; *Ibid*. DDQ00783(GMB1221), living culture, GMBCC1069, GenBank accession number SSU: ON124716; *Ibid*. DDQ00792 (GMB1223), living culture, GMBCC1071, GenBank accession number SSU: ON124717; *Ibid*. DDQ00859 (GMB1248), living culture, GMBCC1080, GenBank accession number SSU: OM891824.

Notes: Newly generated strains, GMBCC1056, GMBCC1065, GMBCC1069, GMBCC1071 and GMBCC1080, and specimen GMB1219 grouped together as the sister clade to *R. verrucispora* (CBS 125434, ex-type) with high statistical supports (100% MLBP, 1.00 BYPP) (Figure 1). A comparison of nucleotide base pairs of ITS, *tef1* and *rpb2,* shows that *R. multiloculate* differs from *R. verrucispora*. Considering ITS, *tef1* and *rpb2* loci, *R. multiloculate* differs from *R. verrucispora* in 24/520 (4.6%), 27/1045 (2.6%), and 20/924 bp (2%), respectively. However, the asexual morph of *R. verrucispora* is undetermined yet for the morphological comparison [1]. *Roussoella multiloculate* shares similar morphological features to *R. chiangraina*, *R. neopustulans*, *R. pustulans,* and *R. siamensis* [1], but phylogenetic reconstructions strongly supported that these species are phylogenetically distinct (Figure 1). Therefore, based on the phylogenetic analyses and guidelines provided by Jeewon and Hyde [38] for the delimitation of new species, we introduce *R. multiloculate* as a novel species of *Roussoella*.

***Roussoella nitidula*** Sacc. and Paol., Atti Inst. Veneto Sci. lett., ed Arti, Sér. 6 6:410 (1888) Figure 6.

Index Fungorum number: IF 177454.

Descriptions of sexual and asexual morphs see Liu et al. [1].

Distributions: China, Malaysia, Thailand.

Material examined: China, Yunnan Province, Dehong, Ruili, Yinlong Village (97°55′40.6″ N, 24°15′49.3″ E, 912 m), on dead culms of bamboo, 16 August 2020, Dong-Qin Dai, DDQ00957 (GMB1270); living culture, GMBCC1097 (new country record), GenBank accession number SSU: OM891826.

Notes: In our phylogenetic analyses, the new strain GMBCC1097 grouped with two strains of *R. nitidula* (MFLUCC 11-0182, ex-type) and MFLUCC 11-0634) (Figure 1) with high statistical supports (100% MLBS, 1.00 BYPP). Furthermore, GMBCC1097 morphologically resembles *R. nitidula*, having black dome-shaped ascostromata; hypha-like, septate, numerous, narrow pseudoparaphyses; cylindrical, relatively thin-walled asci and two-celled dark brown ornamented ascospores [1]. Our fresh collections are morphologically similar to the type of *R. nitidula* and have approximate sizes of asci and ascospore (asci 90–130 × 8–11 μm vs. 110–150 × 8–10(–11) μm; ascospores 14.5–17 × 5–6.5 μm vs. 16–18 × 6–7 μm). Therefore, we consider them to represent one species based on phylogeny and morphology. We herein report *R. nitidula* from China for the first time.

***Roussoella papillate*** Dai and Wijayaw. sp. nov. Figure 7.

Index Fungorum number: IF 556010.

Etymology: References to its papilate ascostromata.

Holptype: GMB1298.

*Saprobic* on decaying bamboo culms. Sexual morph: *Ascostromata* 250–350 μm high, 500–900 μm long, 500–700 μm wide, deeply immersed under a brown area, becoming slightly raised at maturity, ellipsoidal to irregular coriaceous, solitary to gregarious, elliptical, with a prominent, black papillate, uniloculate. *Locules* 200–300 μm high, 450–500 μm diam., solitary, subglobose, brown to dark brown, with a central ostiole. *Wall of locules* 9–20 μm wide, composed of 1–2 layers of *textura angularis*, thin-walled flattened at the base, light brown to brown. *Hamathecium* comprises 1–2 μm wide, numerous, anastomosing cellular pseudo paraphyses, branching at the apex, smooth-walled, often constrict at the septum, and embedded in a gelatinous matrix. *Asci* 108–125 × 7–10 μm ($\bar{x}$ = 114.1 × 8.3 μm, n = 20), 8-spored, bitunicate, cylindrical, short pedicellate, apically rounded with an ocular chamber (0.5–0.8 μm). *Ascospores* 15–17 × 5.5–7 μm ($\bar{x}$ = 16.4 × 6.2 μm, n = 20), uniseriate, ellipsoidal to broad fusiform, 2-celled, constricted at the septum, brown to dark brown, with longitudinal striations and surrounded by a mucilaginous sheath. Asexual morph: Undetermined.

Culture characters: Ascospores germinating on PDA within 24 h and germ tubes produced from one cell. Colonies slow-growing, 20 mm diam. after 20 days at 28 °C, under 24 h dark, rounded, with even margin, white, cottony from above, and white at the margin, yellowish-brown at the center, ring-like from below.

Distributions: China.

Material examined: China, Yunnan Province, Luoping, Jiulong fall waterfall, on dead culms of bamboo, 29 August 2020, Dong-Qin Dai, DDQ01076 (GMB1298, holotype); ex-type GMBCC1121; *Ibid*. (IFRD500-24 isotype), ex-isotype living culture IFRDCC 3103, GenBank accession number SSU: ON228186.

Notes: *Roussoella papillate* (GMBCC1121) formed a well distinct lineage (100% MLBP, 1.00 BYPP) basal to *R. hysterioides* (CBS 546.94) and *R. japanensis* (MAFF 239636, ex-type, GMBCC1067, and GMBCC1117) (Figure 1). Base pair differences in the *tef1* gene region of *R. papillate* to *R. hysterioides* and *R. japanensis* are 4/841 (0.5%) and 18/923 bp (2%), respectively, while base pair differences of ITS locus of these three species are very less. Morphological differences between *R. papillate* and related species are listed in Table 5. Therefore, depending on morphological differences and slight base pair differences in *tef1* region, we introduce this species as a new member of *Roussoella*.

***Roussoella padinae*** Prigione, Bovio, Poli, and Varese, Diversity 12(4, no. 144):14 (2020) Figure 8.

Index Fungorum number: IF 832843.

*Saprobic* on dead bamboo culms. Sexual morph: Undetermined. Asexual morph: *Stromata* forming under a light red area, up to 1–1.5 mm diam. and 100–200 µm high, and becoming raised when mature, ellipsoidal to globose. *Conidiomata* 85–175 µm high, 200–980 µm diam., loculate, 2–5 locules gregarious immersed in the stromata, fabiform, dark brown. *Conidiomatal wall* comprising two layers of cells of *textura angularis*, with dark to brown outer thin layer 3 µm thick, with 6.3–7.5 µm thick, hyaline, conidiogenous inner layer. *Conidiophores* were reduced to conidiogenous cells. *Conidiogenous cells* 1.5–4.5 × 1–2.5 µm ($\bar{x}$ = 2.7 × 1.6 µm, n = 20) enteroblastic, indeterminate, discrete, cylindrical to ampulliform, hyaline, smooth-walled. *Conidia* 3–4 × 2–3 µm ($\bar{x}$ = 3.3 ×2.4 µm, n = 20), ellipsoidal to globose, aseptate, straight, rounded at both ends, hyaline when young, and becoming brown at maturity, smooth-walled, inside usually containing one small guttule.

Culture characters: Conidia germinating on PDA within 24 h and germ tubes produced from one side. Colonies slow-growing, 40 mm diam. after 20 days at 28 °C, under 24 h dark, circular, with radialized margin, floccose at the center, brown from above and dark brown from below.

Distributions: China, Italy.

Material examined: China, Yunnan Province, Jinghong Menla, Manzhang (21°91′97.56″ N, 101°20′42.49″ E, 617.14 m), on dead culms of bamboo, 16 July 2020, Dong-Qin Dai, DDQ02019, GMB1320; living culture, GMBCC1126.

Notes: Phylogenetic analyses showed that the new strain GMBCC1126 grouped with *R. padinae* (MUT 5503, ex-type) which was introduced by Poli et al. [8] from brown alga *Padina pavonica* (i.e., from marine habitat). Base pair arrangement of ITS, LSU, and *rpb2* loci are identical in strain GMBCC1126 and *R. padinae*. Therefore, GMB1320 (strain: GMBCC1126) and *R. padinae* are conspecific. However, sexual and asexual morphs of *R. padinae* are undetermined while Poli et al. [8] have described this species based only on the morphology of colonies and vegetative structures. In this study, we report the asexual morph of *R. padinae* for the first time from dead culms of bamboo. Further, this is the first record of this species from terrestrial habitats and the first record from China.

***Roussoella scabrispora*** (Höhn.) Aptroot, Nova Hedwigia 60(3-4):368 (1995) Figure 9.

Index Fungorum number: IF 414110.

Description of sexual morph see Liu et al. [1], Asexual morph: Undetermined.

Distributions: China, Indonesia, Thailand.

Material examined: China, Yunnan Province, Ruili, Dehong Yinlong Village (97°55′40.6″ N, 24°15′49.3″ E, 912 m), on dead culms of bamboo, 16 August 2020, Dong-Qin Dai, DDQ00960 (GMB1274); living culture, GMBCC1101, GenBank accession number SSU: OM891827; *Ibid*. DDQ00961 (GMB1275), GenBank accession number SSU: ON124718; living culture, GMBCC1102; *Ibid*. DDQ01003 (GMB1286); living culture, GMBCC1108, GenBank accession number SSU: ON124719; Bamboo Garden, Xishuangbanna Topic Botany Garden, Mengla, Jinghong, Yunnan, China (101°24′41.74″ N, 21°93′40.94″ E, 507.88 m), on dead culms of bamboo, 15 August 2020, Dong-Qin Dai, DDQ00975 (GMB1279); living culture, GMBCC1104 (new country record), GenBank accession number SSU: OM891828.

Notes: *Roussoella scabrispora* comprises distinctive ascospores with reticulate wall ornamentation [1]. However, the ascospores of our new collection are slightly narrower (27–34 × 7–9.5 μm vs. (24−)25–29(−32) × (7−)9–10.5 μm) than in the protologue [39]. Nevertheless, based on phylogenetic analyses (Figure 1), we confirmed new collections (i.e., GMB1274, GMB1286 and GMB1279) are *R. scabrispora*. This is the first report of *R. scabrispora* from China.

***Roussoella sinensis*** Dai and Wijayaw. sp. nov. Figure 10.

Index Fungorum number: IF 556014.

Etymology: Reference to its first collection site China.

Holotype: GMB1296.

*Saprobic* on decaying bamboo culms. Sexual morph: *Ascostromata* 400–600 μm diam., forming under raised, visible, dark brown to black, globose areas near ostioles opening 100–200 diam. on host surface, deeply immersed, scattered to gregarious, uniloculate. *Locules* 160–200 μm high, 250–400 μm diam., subglobose, dark brown, with a short central ostiolate neck, 55–65 μm high, 40–45 μm diam. *Wall of locules* 10–20 μm thick, comprising host and fungal tissues, thin, 8–13 μm wide, composed of dark brown cells of *textura angularis*. *Hamathecium* comprises 1–2 μm wide, numerous, anastomosing branched pseudoparaphyses, rough-walled, and embedded in a gelatinous matrix. *Asci* 130–170 μm × 9–10.5 μm ($\bar{x}$ = 148.9 × 9.6 μm, n = 20), 8-spored, bitunicate, cylindrical, with a short knob-like pedicel, with an ocular apical chamber. *Ascospores* 16.5–20.5 × 6–7.5μm ($\bar{x}$ = 18.2 × 6.8 μm, n = 20), uniseriate, overlapping, ellipsoid to broad fusiform, 2-celled, upper cells bigger, occasionally curve, brown, constricted at the septum, narrowly at both ends, with longitudinal striations and surrounded by a mucilaginous sheath. Asexual morph: Undetermined.

Culture characters: Ascospores germinating on PDA within 24 h and germ tubes produced from upper cell. Colonies rapidly growing, 20 mm diam. after 20 days at 28 °C, under 24 h dark, ellipsoidal to rounded, with even, thallus-like margin, pale yellow, floccus at the margin, milk-white at the center and becoming light brown.

Distributions: China.

Material examined: China, Yunnan Province, Jinghong, Mengla, Bamboo Garden, Xishuangbanna Topic Botany Garden (101°24′41.74″ N, 21°93′40.94″ E, 507.88 m), on dead culms of bamboo, 15 August 2020, Dong-Qin Dai, DDQ01045 (GMB1296, holotype); ex-type GMBCC1119, GenBank accession number SSU: OM891833; *Ibid*. (IFRD500-22, isotype), ex-isotype IFRDCC 3101, GenBank accession number SSU: ON228185.

Notes: *Roussoella sinensis* (GMB1296, holotype) shows the typical morphological characters of the sexual morphs of *Roussoella* species [1,17], and is distinct by having locules with a short central ostiolate neck (Figure 10c), 55–65 μm high, 40–45 μm diam. Furthermore, *Roussoella sinensis* grouped as the sister species to *R. doimaesalongensis* (MFLUCC 14-0584, ex-type). ITS, *tef1* and *rpb2* base pair differences of *R. sinensis* (GMBCC1119) and *R. doimaesalongensis* are 10% (53/530), 21.30% (171/803), and 9.6% (70/728), respectively. *Roussoella*
*siamensis* (MFLUCC 11-0149, ex-type) and *R.*
*yunnanensis* (MFLUCC 18-0115, ex-type) as a separate lineage but with relatively less statistical support (Figure 1). ITS, *tef1,* and *rpb2* base pair differences of *R. sinensis* (GMBCC1119) and *R. siamensis* (MFLUCC 11-0149) are 15.20% (62/515), 18.53% (192/1036), and 6.60% (60/909) respectively. Base pair differences of ITS and *rpb2* genes locus of *Roussoella sinensis* (GMBCC1119) and *R. yunnanensis* (MFLUCC 18-0115) are 15.20% (71/467) and 5.51% (50/907), respectively. However, *tef1* sequences from *R. yunnanensis* (MFLUCC 18-0115) are not available in the GenBank database. *Roussoella doimaesalongensis* was introduced by Thambugala et al. [40] based on a specimen collected on dead bamboo in Thailand. However, only the asexual morph is known for *R. doimaesalongensis* [40]. According to the recommendations suggested by Jeewon and Hyde [38] regarding base pair differences, *Roussoella sinensis* is a phylogenetically different species from *R. doimaesalongensis*, *R.*
*siamensis,* and *R. yunnanensis;* hence, we introduced *Roussoella sinensis* as a new member to *Roussoella*.

***Roussoella tuberculata*** Dai and Hyde, Fungal Diversity 82:37 (2016) Figure 11

Index Fungorum number: IF 552027.

Sexual morph: Undetermined. Description of asexual morph see Dai et al. [17].

Distributions: China, Thailand.

Material examined: China, Yunnan Province, Jinghong, Mengla, Bamboo Garden, Xishuangbanna Topic Botany Garden (101°24′41.74″ N, 21°93′40.94″ E, 507.88 m), on dead culms of bamboo, 15 August 2020, Dong-Qin Dai, DDQ02000 (GMB1317); living culture, GMBCC1123 (new country record), GenBank accession number SSU: OM891834.

Notes: Our fresh collection (GMB1317) shares similar morphologies of *R. tuberculata* which was introduced by Dai et al. [17] from Thailand. *Roussoella tuberculata* is characterized by large, immersed, eustromatic conidiomata, which are rather flattened, phialidic, annellidic conidiogenous cells and conidia covered by small tubercules [17]. Usually, the conidia of *Roussoella* species comprise verrucose wall ornamentation [1,39], while *R. tuberculata* has conidia with small and roughened tubercules. The above morphological identification is confirmed by our phylogenetic analyses and base-pair comparisons as well (100% MLBS, 1.00 BYPP, Figure 1). This is the first record of this species from China.

***Roussoella uniloculata*** Dai and Wijayaw., sp. nov. Figure 12.

Index Fungorum number: IF 556016.

Etymology: Uniloculata means single locule.

Holotype: GMB1288.

*Saprobic* on decaying bamboo culms. Sexual morph: *Ascostromata* 250–350 μm diam., forming under raised, visible, black, round areas on host surface and becoming prominent when maturity, containing a single locule. *Locules* 120–145 μm high, 270–380 μm diam., solitary, subglobose to ellipsoidal, brown, with an inconspicuous central ostiole. *Wall of locules* comprising host and fungal tissues, thin, 7–10 μm wide, composed of dark brown cells of *textura angularis*. *Hamathecium* comprises 1–2.5 μm wide, numerous, anastomosing branched cellular pseudoparaphyses, rough-walled, and embedded in a gelatinous matrix. *Asci* 61.5–102.5 × 4–6.5 μm ($\bar{x}$ = 78.8 × 5.2 μm, n = 20), 6–8-spored, bitunicate, cylindrical, with a short knob-like pedicel, with an ocular apical chamber. *Ascospores* 8.5–12 × 3.5–4.5(−4.8) μm ($\bar{x}$ = 10.3 × 4.1 μm, n = 20), uniseriate, ellipsoid to broad fusiform, 2-celled, upper cells bigger, occasionally curved, brown, constricted at the septum, with longitudinal striations and surrounded by a mucilaginous sheath. Asexual morph: Undetermined.

Culture characters: Ascospores germinating on PDA within 24 h and germ tubes produced from both cells. Colonies slow-growing, 30 mm diam. after 20 days at 28 °C, under 24 h dark, elliptic to round, with irregular margin, milk-white at margin, and light brown to dark brown at the central.

Distributions: China.

Material examined: China, Yunnan Province, Ruili, Dehong Yinlong Village (97°55′40.6″ N, 24°15′49.3″ E, 912 m), on dead culms of bamboo, 16 August 2020, Dong-Qin Dai, DDQ01005 (holotype GMB1288), ex-type GMBCC1110, GenBank accession number SSU: OM891829; *Ibid*. (IFRD500-23 isotype), ex-isotype IFRDCC 3102; *Ibid*. DDQ01005-2, GenBank accession number SSU: OM891835.

Notes: *Roussoella uniloculata* (GMB1288) formed a basal lineage in a clade comprising *R. angusta* (MFLUCC 15-0186, ex-type), *R. chiangraina* (MFLUCC 10-0556), *R*. *kunmingensis* (KUMCC 18-0128), *R. magnatum* (MFLUCC 15-0185), *R*. *mediterranea* (MUT 5369), *R*. *neopustulans* (MFLUCC 11-0609), and *R*. *padinae* (MUT 5503) with high statistical supports (100% MLBP, 1.00 BYPP) (Figure 1). Base pair differences of ITS and *rpb2* gene loci of *R. uniloculata* and other related species showed that they are phylogenetically distinct species [38] (Table 6).

*Roussoella uniloculata* is similar to *R. yunnanensis* and *R. siamensis* in having ellipsoidal to fusiform and 2-celled ascospores with longitudinal striations. However, *R. uniloculata* can be distinguished from *R. yunnanensis* in having smaller ascostromata (250–350 μm diam. vs. 1–1.3 mm diam.) with single locule vs. with multiple locules, and asci with a short knob-like pedicel vs. asci with a slightly furcate short pedicel [5]. *Roussoella uniloculata* can be distinguished from *R. siamensis* in having smaller ascostromata (250–350 μm diam. vs. 620–750 μm diam. [17], higher locules (120–145 μm high vs. 70–120 μm high; [17]. *Roussoella uniloculata* can be compared with *R. pustulans*, in having small ascospores with bigger upper cells (8.5–12 × 3.5–4.5 μm vs. 10–16 × 4–5 μm) [1]. However, *R. uniloculata* differs by smaller ascostromata (165–300 μm diam. vs. 1 mm diam.), and in the phylogenetic tree, they form separate lineages (Figure 1). Moreover, morphology, host, and distribution of the new species were compared with the known species which are lacking sequence data (Table 7). Hence, based on molecular phylogenetic and morphological analyses, we introduced *R. uniloculata* as a novel species of *Roussoella*.

## 4. Discussion

The family *Roussoellaceae* (in *Pleosporales*), comprises the genera *Neoroussoella*, *Roussoella* and *Roussoellopsis* [1], which are saprobes in different hosts, especially bamboo and palms (terrestrial and aquatic environments) or human pathogens [46]. Recently, three new species have been isolated from marine environments [8]. Currently, the family comprises 12 genera reported as sexual, asexual, or holomorph [1,8,9,17,46,47]. This study introduced four new species of *Roussoella*: three reported as sexual morphs (i.e., *R. papillate*, *R. sinensis,* and *R. uniloculata*), and another as an asexual morph (i.e., *R. multiloculate*). All species have been reported as saprobes of bamboo plants.

However, asexual morphs of *R. kunmingensis* (described initially as a sexual taxon [8]) and *R. padinae* (described initially as without sexual or asexual characteristics fide [8]) were reported for the first time in the present study. Sexual and asexual links were established based on DNA sequence analyses (Figure 1). The sexual morph of *R**. kunmingensis* was also reported from Kunming, Yunnan Province. Nevertheless, Poli et al. [8] introduced *R. padinae* from brown alga *Padina pavonica* (Italy), and thus from the marine environment. This finding indicates that *Roussoella* species present a broad range of habitats and distribution. Further, the same species could be reported from different environments but as its alternative morph.

*Neoroussoella bambusae* (Thailand [1]), *R. japanensis* (Japan [1]), *R. nitidula* (Malaysia [1]), *R. scabrispora* (Indonesia [41]), and *R. tuberculata* (Thailand) species were reported from China for the first time. These records confirmed that the members of *Roussoellaceae* have a broad range of geographical distribution in Southeast Asia and Central Asia. We predict that novel species could occur in other tropical Asian countries, such as India, Laos, Myanmar, Pakistan, and Sri Lanka.

The genus *Roussoella* comprises 51 epithets [47], but only 46 species are listed in Species Fungorum [47]. Eight species were transferred to *Dothideaceae*, *Thyridariaceae*, *Phyllachoraceae,* and *Diaporthales* [2,7,28,48]. Further, two species were transferred to *Neoroussoella* and *Pseudoroussoella*, which are also nested in *Roussoellaceae* [49]. Moreover, *R. hysterioides* var. *minuta* (Hino and Katum) Hino and Katum were listed as synonyms of *R. hysterioides* in Index Fungorum [47], and two species *R. phyllostachydis* and *R. minutella* were synonymized as *R. pustulans* by Hyde [41]. Thus, a total of 33 taxa were accepted in *Roussoellaceae*. Hongsanan et al. [46] mentioned that DNA sequence data are available only for 22 species. Hence, it is essential to recollect known species lacking DNA sequence data and designate epitypes. In this study, we compared the morphological characters of the sequence lacking *Roussoella* taxa [12,41,42,43,44,45,50] prior to introducing the new taxa (Table 7).

## Figures and Tables

**Figure 1 jof-08-00532-f001:**
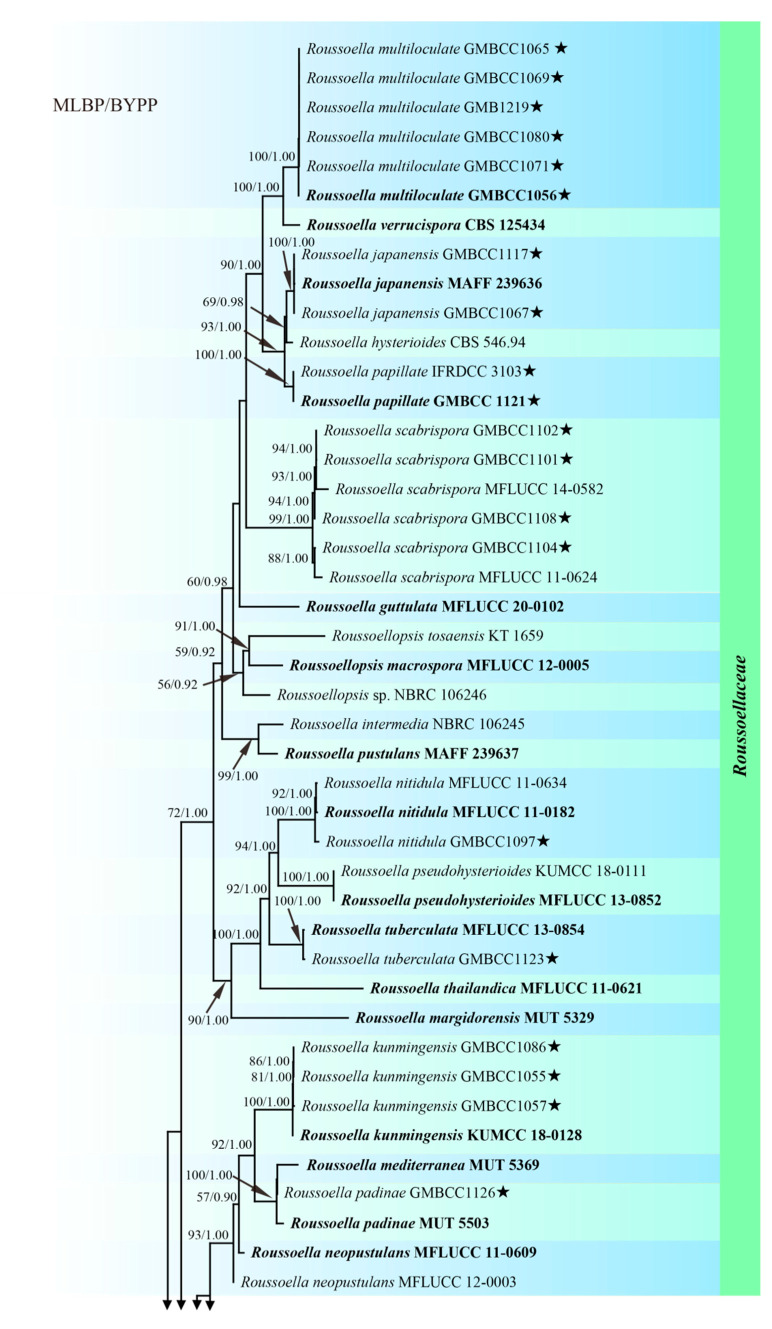
Phylogenetic tree from the best scoring of the RAxML analysis based on combined ITS, LSU, *rpb2* and *tef1* loci is rooted to *Torula herbarum* (CBS 111855) and *T. hollandica* (CBS 220.69). Bootstrap values for maximum likelihood (MLBP) and Bayesian posterior probabilities (BYPP) equal to or greater than 50% and 0.80, respectively, are given at the branches. The newly generated sequences are marked with asterisk “★” and ex-type strains are indicated in bold. Bar = 0.1 expected number of nucleotide substitutions per site per branch.

**Figure 2 jof-08-00532-f002:**
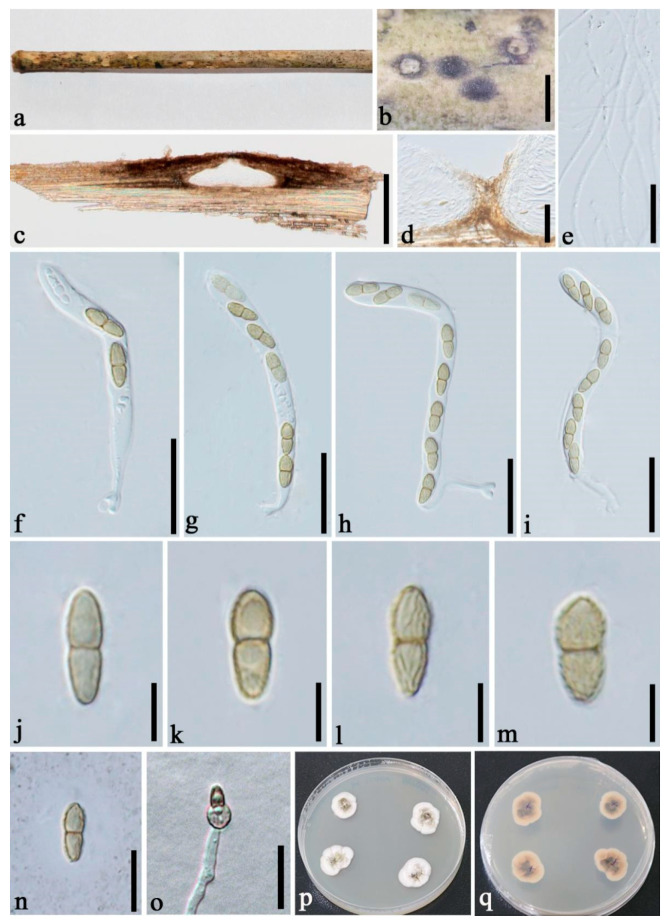
*Neoroussoella bambusae* (GMB1291, new country record). (**a**) Bamboo specimen; (**b**) Black ascostromata on host surface; (**c**) Vertical section of ascostroma; (**d**) Cells of locules walls; (**e**) Pseudoparaphyses; (**f**–**i**) Asci; (**j**–**m**) Ascospores; (**n**) Ascospore in India ink; (**o**) Germinating ascospore; (**p**,**q**) Cultures on PDA from above and below. Scale bars: (**b**) = 150 μm, (**c**) = 100 μm, (**d**) = 50 μm, (**e**,**n**) = 10 μm, (**f**–**i**) = 20 μm, (**j**–**m**) = 5 μm, (**o**) = 15 μm.

**Figure 3 jof-08-00532-f003:**
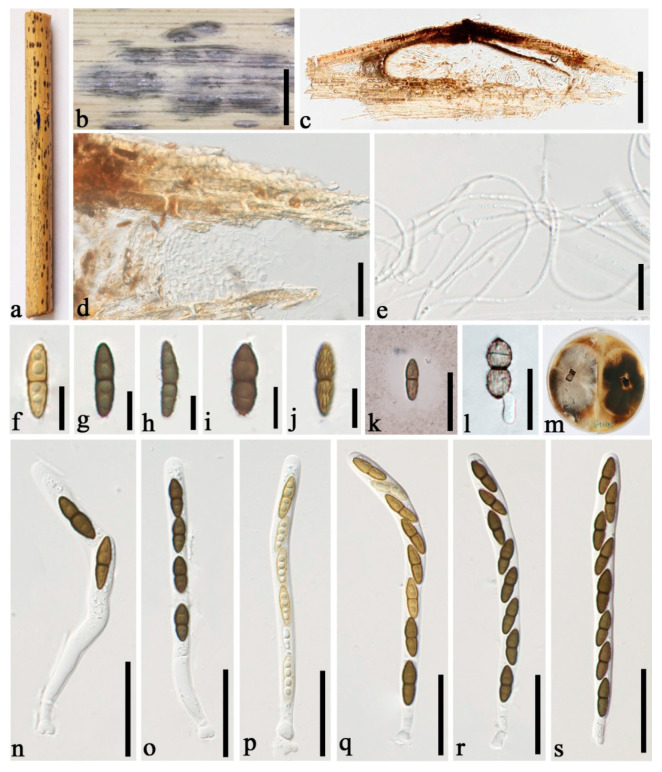
*Roussoella japanensis* (GMB1292, new country record). (**a**) Bamboo specimen; (**b**) black ascostromata on host surface; (**c**) vertical section of ascoma; (**d**) cells between locules; (**e**) pseudoparaphyses; (**f**–**j**) ascospores; (**k**) ascospore in India ink; (**l**) germinating ascospore; (**m**) cultures on PDA from below and above; (**n**–**s**) asci. Scale bars: (**b**) = 2 mm, (**c**) = 200 μm, (**d**) = 50 μm, (**e**–**j**) = 10 μm, (**k**,**l**) = 20 μm, (**n**–**s**) = 30 μm.

**Figure 4 jof-08-00532-f004:**
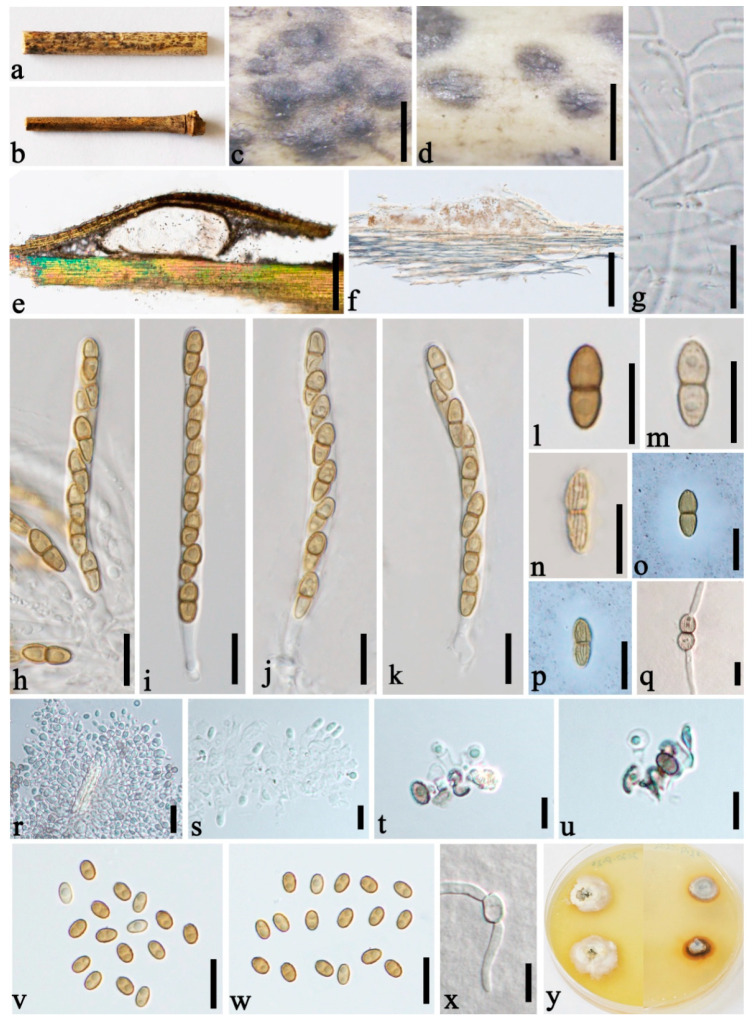
*Roussoella kunmingensis* (GMB1203, sexual morph, (**a**,**c**,**e**,**g**,**h**–**q**); GMB1259, first report of asexual morph, (**b**,**d**,**f**,**r**–**y**)). (**a**,**b**) Bamboo specimens; (**c**) black ascostromata on host surface; (**d**) black conidiomata on host surface; (**e**) vertical section of ascostromata; (**f**) vertical section of conidioma; (**g**) pseudoparaphyses; (**h**–**k**) asci; (**l**–**n**) ascospores; (**o**,**p**) ascospores in India ink; (**q**) germinating ascospore; (**r**–**u**) conidiogenous cells contacting with conidia; (**v**,**w**) conidia; (**x**) germinating conidium; (**y**) cultures on PDA from above and below. Scale bars: (**c**) = 500 μm, (**d**) = 300 μm, (**e**) = 200 μm, (**f**) = 100 μm, (**g**–**x**) = 10 μm.

**Figure 5 jof-08-00532-f005:**
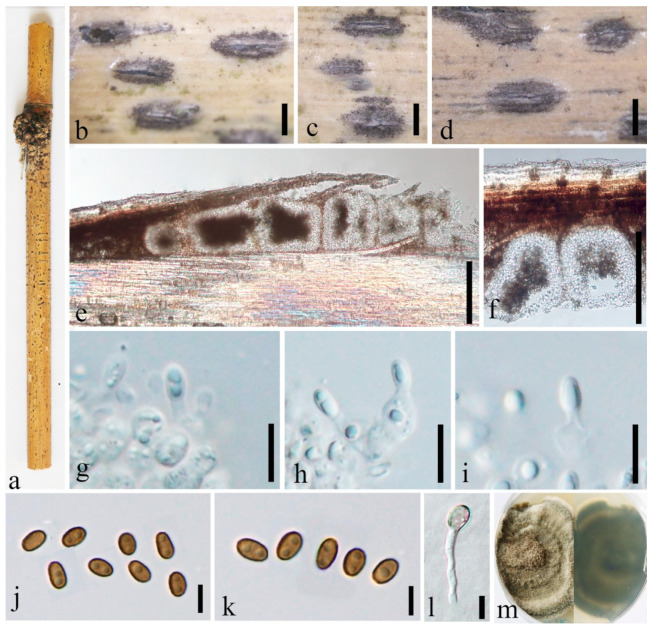
*Roussoella multiloculate* (GMB1207, holotype). (**a**) Bamboo specimen; (**b**–**d**) black conidiomata on host surface; (**e**,**f**) vertical sections of conidiomata; (**g**–**i**) conidia attached to conidiogenous cells; (**j**,**k**) conidia. (**l**) germinating conidium. (**m**) cultures on PDA from above and below. Scale bars: (**b**–**d**) = 500 µm, (**e**,**f**) = 100 µm, (**g**–**l**) = 5 µm.

**Figure 6 jof-08-00532-f006:**
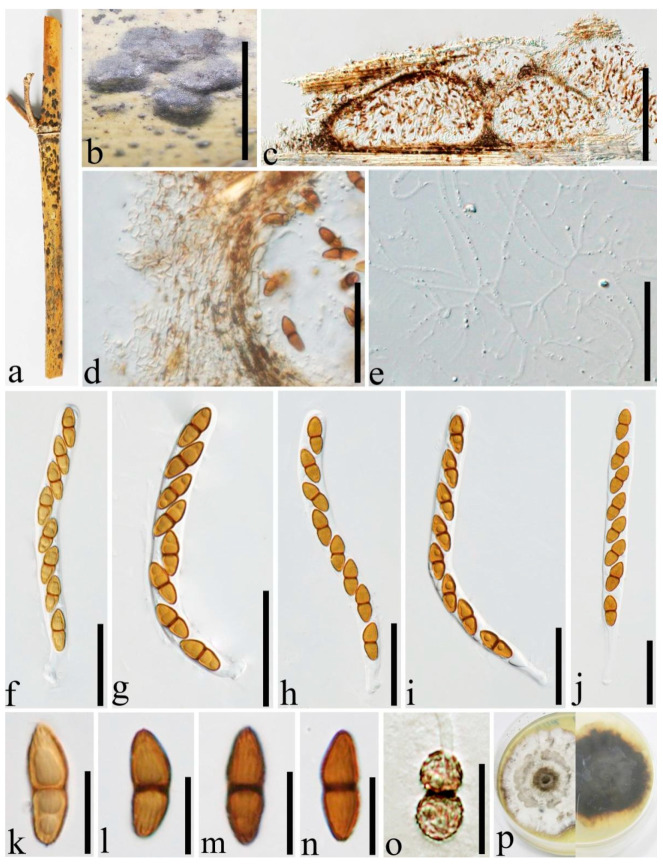
*Roussoella nitidula* (GMB1270, new country record). (**a**) Bamboo specimen; (**b**) black ascostromata on host surface; (**c**) vertical sections of ascomata; (**d**) cells of locule wall. (**e**) pseudoparaphyses; (**f**–**j**) asci; (**k**–**n**) ascospores; (**o**) Germinating ascospore; (**p**) Cultures on PDA from above and below. Scale bars: (**b**) = 1 mm, (**c**) = 500 μm, (**d**) = 50 μm, (**e**–**j**) = 30 μm, (**k**–**n**) = 10 μm, (**o**) = 15 μm.

**Figure 7 jof-08-00532-f007:**
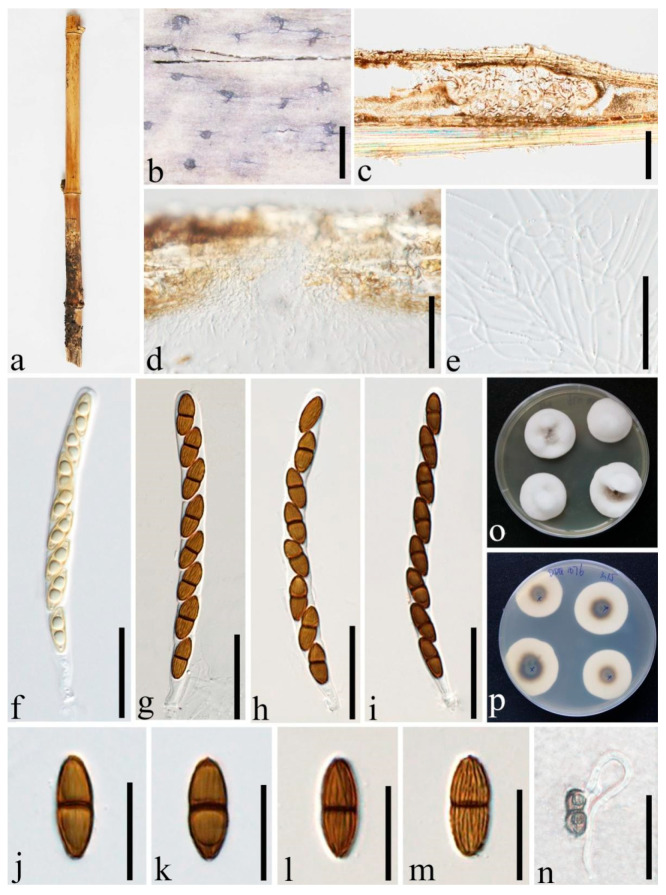
*Roussoella papillate* (GMB129, holotype). (**a**) Bamboo specimen; (**b**) black ascostromata on host surface; (**c**) vertical section of ascoma; (**d**) cells of locule wall near the ostiole; (**e**) branching pseudoparaphyses; (**f**–**i**) asci; (**j**–**m**) ascospores; (**n**) germinating ascospore; (**o**,**p**) cultures on PDA from above and below. Scale bars: (**b**) = 500 µm, (**c**) = 150 µm, (**d**) = 50 µm, (**e**–**i**,**n**) = 30 µm, (**j**–**m**) = 15 µm.

**Figure 8 jof-08-00532-f008:**
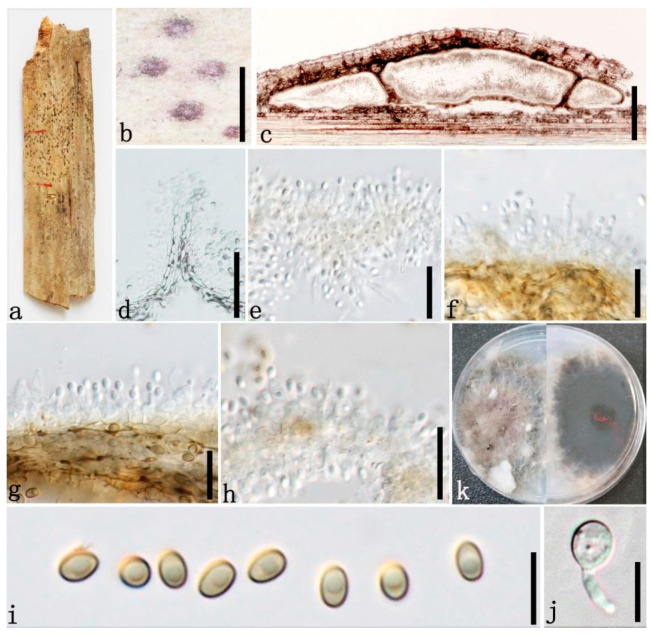
*Roussoella padinae* (GMB1320, first report of asexual morph, first record from terrestrial habitat and first record from China). (**a**) Bamboo specimen; (**b**) reddish-brown conidiomata on host surface; (**c**) vertical sections of conidiomata; (**d**) conidioma wall; (**e**–**h**) conidiogenous cells; (**i**) conidia; (**j**) germinating conidium; (**k**) cultures on PDA from above and below. Scale bars: (**b**) = 1 mm, (**c**) = 150 µm, (**d**) = 30 µm, (**e**–**h**) =15 µm, (**i**) = 5 µm, (**j**) = 10 µm.

**Figure 9 jof-08-00532-f009:**
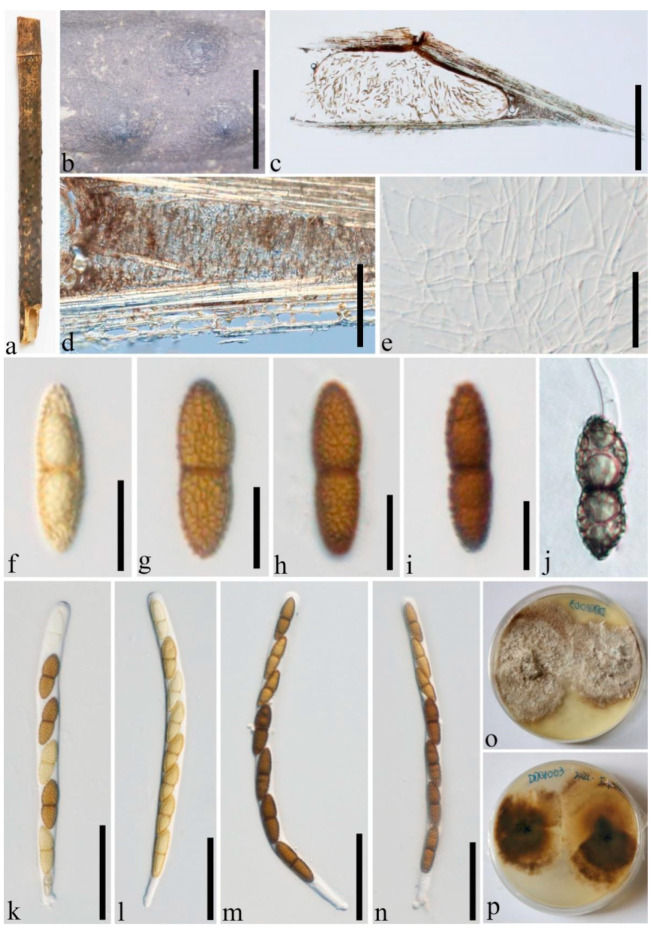
*Roussoella scabrispora* (GMB1286, new country record). (**a**) Bamboo specimen; (**b**) ascomata on bamboo host; (**c**) vertical section of ascoma; (**d**) peridium; (**e**) pseudoparaphyses; (**f**–**i**) ascospores; (**j**) germinating ascospore; (**k**–**n**) asci; (**o**,**p**) cultures on PDA from above and below. Scale bars: (**b**) = 1 mm, (**c**) = 500 μm, (**d**) = 100 μm, (**e**) = 30 μm, (**f**–**j**) = 15 μm, (**k**–**n**) = 50 μm.

**Figure 10 jof-08-00532-f010:**
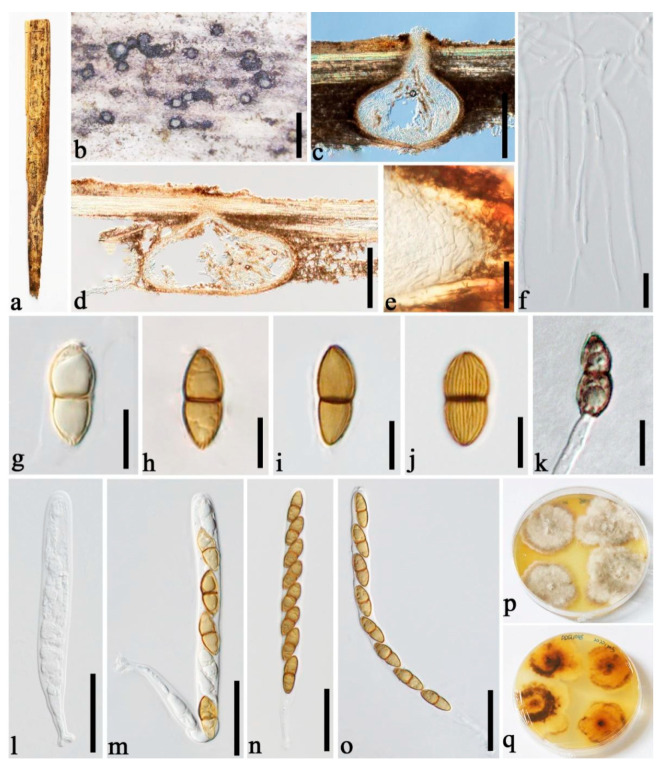
*Roussoella sinensis* (GMB1296, holotype). (**a**) Bamboo specimen; (**b**) black ascostromata showing black ostioles with openings on host surface; (**c**,**d**) vertical section of ascostromata; (**e**) cells of locule wall; (**f**) pseudoparaphyses; (**g**–**j**) ascospores; (**k**) germinating ascospore; (**l**–**o**) different developmental stages of asci; (**p**,**q**) cultures on PDA from above and below. Scale bars: (**c**,**d**) = 200 μm, (**e**) = 50 μm, (**f**) = 10 μm, (**g**–**k**) = 15 μm, (**l**–**o**) = 30 μm.

**Figure 11 jof-08-00532-f011:**
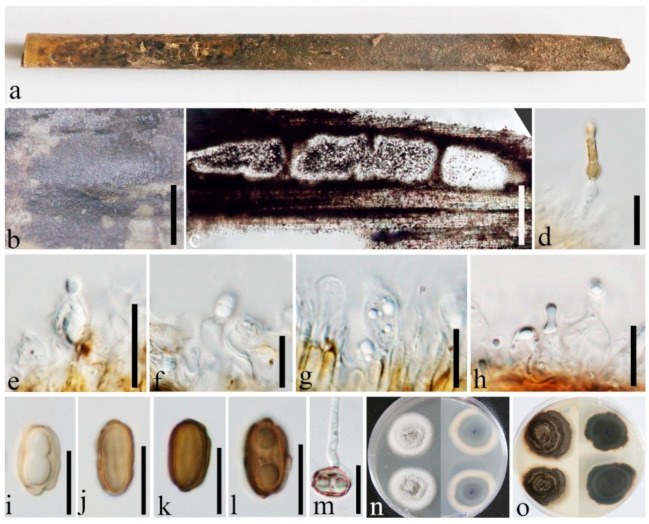
*Roussoella tuberculata* (GMB1317, new country record). (**a**) Bamboo specimen; (**b**) black conidioma on host surface; (**c**) vertical sections of conidiomata; (**d**–**h**) conidiogenous cells and developing conidia; (**i**–**l**) conidia; m: germinating conidium; (**m**) germinating conidium; (**n**) cultures on PDA from above and below after two weeks; (**o**) cultures on PDA from above and below after four weeks. Scale bars: (**b**) = 500 µm, (**c**) = 200 µm, (**d**–**f**,**m**) = 15 µm, (**h**–**l**) = 10 µm.

**Figure 12 jof-08-00532-f012:**
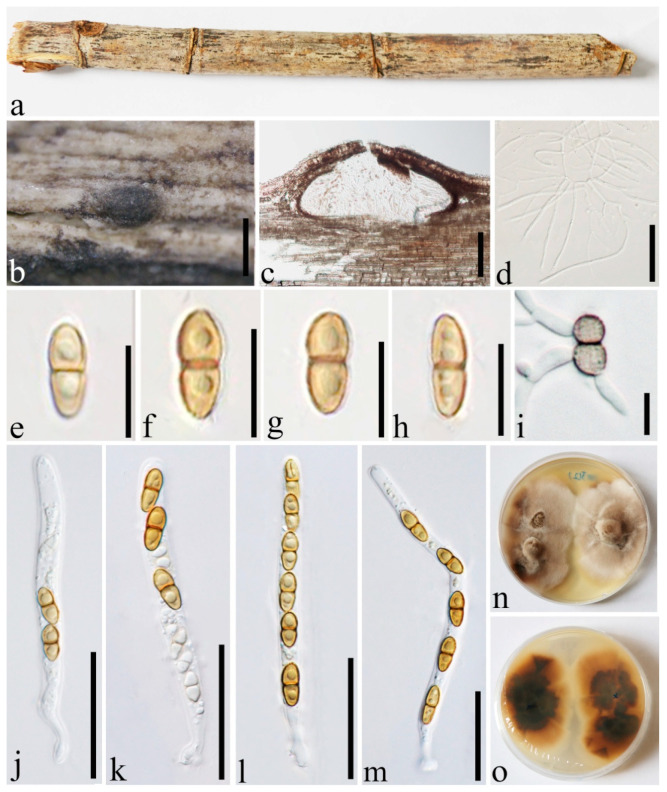
*Roussoella uniloculata* (GMB1288, holotype). (**a**) Bamboo specimen; (**b**) black ascostroma on host surface; (**c**) vertical section of ascomata; (**d**) pseudoparaphyses; (**e**–**h**) ascospores; (**i**) germinating ascospore; (**j**–**m**) asci; (**n**,**o**) cultures on PDA from above and below. Scale bars: (**b**) = 300 µm, (**c**) = 50 µm, (**d**,**j**–**m**) = 30 µm, (**e**–**i**) = 10 µm.

**Table 1 jof-08-00532-t001:** ITS, SSU, LSU, *tef1,* and *rpb2* loci primers information.

Genes	Primers and Base Pairs	References
Internal transcribed spacers (ITS)	Forward: ITS5 TCCTCCGCTTATTGATATGC Reverse: ITS4 GGAAGTAAAAGTCGTAACAAGG	[21]
Large subunit rDNA (LSU)	Forward: LROR GTACCCGCTGAACTTAAGC Reverse: LR5 ATCCTGAGGGAAACTTC	[22]
Small subunit rDNA (SSU)	Forward: NS1 GTAGTCATATGCTTGTCTC Reverse: NS4 CTTCCGTCAATTCCTTTAAG	[21]
Translation elongation factor 1-α gene region (*tef1*)	Forward: EF1-983F GCYCCYGGHCAYCGTGAYTTYAT Reverse: EF1-2218R ATGACACCRACRGCRACRGTYTG	[23]
RNA polymerase II second largest subunit (*rpb2*)	Forward: fRPB2-5f GAYGAYMGWGATCAYTTYGG Reverse: fRPB2-7cr CCCATRGCTTGTYYRCCCAT	[24]

**Table 2 jof-08-00532-t002:** ITS, SSU, LSU, *tef1,* and *rpb2* loci PCR conditions.

Genes	Initial Period	Cycles, Denaturation, Annealing and Elongation	Final Extension	References
ITS, LSU, SSU, *tef1*	94 °C for 3 min	35 cycles of denaturation at 94 °C for 30 s, annealing at 55 °C for 50 s, elongation at 72 °C for 1 min	72 °C for 10 min	[17]
*rpb2*	95 °C for 5 min	40 cycles of denaturation at 95 °C for 1 min, annealing at 52 °C for 2 min, elongation at 72 °C for 90 s	72 °C for 10 min	[17]

**Table 3 jof-08-00532-t003:** Isolates or specimens used in this study and their GenBank accession numbers. The newly generated sequences are marked with asterisk “★” and ex-type strains are in bold, “**-**” means sequences data are unavailable in the GenBank database.

Taxa	Strain/Voucher No.	GenBank Accession Numbers			
		ITS	LSU	*tef1*	*rpb2*
*Arthopyrenia* sp.	UTHSC DI16-362	LT796905	LN907505	LT797145	LT797065
*Arthopyrenia* sp.	UTHSC DI16-334	LT796887	LN907477	LT797127	**-**
** *Neoroussoella alishanense* **	**FU31016**	**MK503816**	**MK503822**	**MK336181**	**MN037756**
** *Neoroussoella bambusae* **	**MFLUCC 11-0124**	**KJ474827**	**KJ474839**	**KJ474848**	**KJ474856**
*Neoroussoella bambusae* ★	GMBCC1116	OM891810	OM884022	ON098358	ON098377
*Neoroussoella bambusae* ★	GMBCC1118	OM891812	OM801294	ON098359	ON098376
*Neoroussoella entadae*	MFLUCC 15-0098	MH275075	MH260309	**-**	**-**
** *Neoroussoella heveae* **	**MFLUCC 17-0338**	**MH590693**	**MH590689**	-	-
*Neoroussoella heveae*	MFLUCC 17-2069	MT310634	MT214589	MT394647	MT394703
** *Neoroussoella lenispora* **	**GZCC 16-0020**	-	**KX791431**	**-**	**-**
** *Neoroussoella leucaenae* **	**MFLUCC 18-1544**	**MK347767**	**MK347984**	**MK360067**	**MK434876**
** *Neoroussoella solani* **	**CPC 26331**	**KX228261**	**KX228312**	**-**	**-**
** *Pararoussoella mangrovei* **	**MFLUCC 16-0424**	**MH025951**	**MH023318**	**MH028246**	**MH028250**
*Pararoussoella mukdahanensis*	KUMCC 18-0121	MH453489	MH453485	MH453478	MH453482
** *Pararoussoella mukdahanensis* **	**MFLUCC 11-0201**	**KU940129**	**KU863118**	**-**	**-**
** *Pararoussoella rosarum* **	**MFLUCC 17-0796**	**NR_157529**	**NG059872**	**MG829224**	**-**
*Parathyridaria percutanea*	CBS 128203	KF322117	KF366448	KF407988	KF366453
** *Parathyridaria percutanea* **	**CBS 868.95**	**KF322118**	**KF366449**	**KF407987**	**KF366452**
** *Parathyridaria ramulicola* **	**CBS 141479**	**KX650565**	**KX650565**	**KX650536**	**KX650584**
*Parathyridaria ramulicola*	MF4	KX650564	KX650564	KX650535	**-**
** *Parathyridaria robiniae* **	**MFLUCC 14-1119**	**KY511142**	**KY511141**	**KY549682**	**-**
** *Pseudoneoconiothyrium euonymi* **	**CBS 143426**	**MH107915**	**MH107961**	**-**	**MH108007**
** *Pseudoneoconiothyrium rosae* **	**MFLUCC 15-0052**	**NR_157523**	**NG059868**	**-**	**-**
** *Pseudoroussoella chromolaenae* **	**MFLUCC 17-1492**	**MT214345**	**MT214439**	**MT235769**	**-**
*Pseudoroussoella elaeicola*	MFLUCC 17-1483	MT214348	MT214442	MT235772	MT235808
*Pseudoroussoella elaeicola*	MFLUCC 15-0276b	MH742330	MH742327	**-**	**-**
** *Pseudoroussoella elaeicola* **	**MFLUCC 15-0276a**	**MH742329**	**MH742326**	**-**	**-**
** *Roussoella angusta* **	**MFLUCC 15-0186**	**-**	**KT281979**	**-**	**-**
** *Roussoella aquatica* **	**MFLUCC 18-1040**	NR171975	**NG073797**	**-**	**-**
** *Roussoella arundinacea* **	**CBS 146088**	MT223838	**MT223928**	MT223723	**-**
** *Roussoella chiangraina* **	**MFLUCC 10-0556**	**KJ474828**	**KJ474840**	**KJ474849**	**KJ474857**
** *Roussoella doimaesalongensis* **	**MFLUCC 14-0584**	**KY026584**	**KY000659**	**KY651249**	**KY678394**
*Roussoella elaeicola*	MFLUCC 15-0276a	MH742329	MH742326	**-**	**-**
*Roussoella elaeicola*	MFLUCC 15-0276b	MH742330	MH742327	**-**	**-**
** *Roussoella guttulata* **	**MFLUCC 20-0102**	**NR_172428**	**NG_075383**	MW022188	MW022187
*Roussoella hysterioides*	CBS 546.94	KF443405	KF443381	KF443399	KF443392
*Roussoella intermedia*	NBRC 106245	KJ474831	AB524624	**-**	**-**
*Roussoella intermedia*	CBS 170.96	KF443407	KF443382	KF443398	KF443394
** *Roussoella japanensis* **	**MAFF 239636**	**KJ474829**	**AB524621**	**AB539114**	**AB539101**
*Roussoella japanensis* ★	GMBCC1067	OM891802	OM884018	ON098344	ON098381
*Roussoella japanensis* ★	GMBCC1117	OM891811	OM884023	ON098345	ON098382
** *Roussoella kunmingensis* **	**KUMCC 18-0128**	**MH453491**	**MH453487**	**MH453480**	**MH453484**
*Roussoella kunmingensis* ★	GMBCC1055	OM891797	OM884013	ON098353	ON098385
*Roussoella kunmingensis* ★	GMBCC1057	OM891798	OM884014	ON098354	ON098362
*Roussoella kunmingensis* ★	GMBCC1086	OM891804	OM801287	ON098355	ON098363
** *Roussoella magnatum* **	**MFLUCC 15-0185**	**-**	**KT281980**	**-**	**-**
** *Roussoella margidorensis* **	**MUT 5329**	**KU314944**	**MN556322**	**MN605897**	**MN605917**
** *Roussoella mediterranea* **	**MUT 5369**	**KU314947**	**MN556324**	**MN605899**	**MN605919**
** *Roussoella mexicana* **	**CPC 25355**	**KT950848**	**KT950862**	**-**	**-**
*Roussoella multiloculate* ★	GMB1219	OM891801	OM884017	ON098341	ON098366
** *Roussoella multiloculate* ** **★**	**GMBCC1056**	**OM891799**	**OM884015**	**ON098343**	**ON098369**
*Roussoella multiloculate* ★	GMBCC1065	OM891800	OM884016	ON098338	ON098364
*Roussoella multiloculate* ★	GMBCC1069	OM891803	OM884019	ON098340	ON098365
*Roussoella multiloculate* ★	GMBCC1071	ON159383	OM755586	ON098342	ON098368
*Roussoella multiloculate* ★	GMBCC1080	ON159384	OM755589	ON098339	ON098367
** *Roussoella neopustulans* **	**MFLUCC 11-0609**	**KJ474833**	**KJ474841**	**KJ474850**	**-**
*Roussoella neopustulans*	MFLUCC 12-0003	KU940130	KU863119	-	-
** *Roussoella nitidula* **	**MFLUCC 11-0182**	**KJ474835**	**KJ474843**	**KJ474852**	**KJ474859**
*Roussoella nitidula*	MFLUCC 11-0634	KJ474834	KJ474842	KJ474851	KJ474858
*Roussoella nitidula* ★	GMBCC1097	OM891805	OM884020	ON098351	ON098384
** *Roussoella padinae* **	**MUT 5503**	**KU158170**	**MN556327**	**MN605902**	**MN605922**
*Roussoella padinae* ★	GMBCC1126	OM891816	OM884025	ON098356	ON098383
** *Roussoella papillate* ** **★**	**GMBCC1121**	**OM891814**	**OM755608**	**ON098346**	**ON098378**
*Roussoella papillate* ★	IFRDCC 3103	ON228188	ON228184	ON244452	ON244450
** *Roussoella pseudohysterioides* **	**MFLUCC 13-0852**	**KU940131**	**KU863120**	**KU940198**	**-**
*Roussoella pseudohysterioides*	KUMCC 18-0111	MH453490	MH453486	MH453479	MH453483
** *Roussoella pustulans* **	**MAFF 239637**	**KJ474830**	**AB524623**	**AB539116**	**AB539103**
** *Roussoella scabrispora* **	**MFLUCC 11-0624**	**KJ474836**	**KJ474844**	**KJ474853**	**KJ474860**
*Roussoella scabrispora*	MFLUCC 14-0582	KY026583	KY000660	**-**	**-**
*Roussoella scabrispora* ★	GMBCC1101	ON159385	OM755615	ON098347	ON098371
*Roussoella scabrispora* ★	GMBCC1102	OM891806	OM884021	ON098348	ON098370
*Roussoella scabrispora* ★	GMBCC1104	OM891807	OM755616	ON098349	ON098373
*Roussoella scabrispora* ★	GMBCC1108	OM891808	OM755614	ON098350	ON098372
** *Roussoella siamensis* **	**MFLUCC 11-0149**	**KJ474837**	**KJ474845**	**KJ474854**	**KJ474861**
** *Roussoella sinensis* ** **★**	**GMBCC1119**	**OM891813**	**OM884024**	**ON098357**	**ON098379**
*Roussoella sinensis* ★	IFRDCC 3101	ON228187	ON228183	ON244453	ON244451
** *Roussoella thailandica* **	**MFLUCC 11-0621**	**KJ474838**	**KJ474846**	**-**	**-**
** *Roussoella tuberculata* **	**MFLUCC 13-0854**	**KU940132**	**KU863121**	**KU940199**	**-**
*Roussoella tuberculata* ★	GMBCC1123	OM891815	OM755613	ON098352	ON098380
** *Roussoella uniloculata* ** **★**	**GMBCC1110**	**OM891809**	**OM801286**	**ON098360**	**ON098374**
*Roussoella uniloculata* ★	DDQ01005-2	OM891817	OM884026	ON098361	ON098375
** *Roussoella verrucispora* **	**CBS 125434**	**KJ474832**	**AB524622**	**AB539115**	**AB539102**
** *Roussoella yunnanensis* **	**KUMCC 18-0115**	**MH453492**	**MH453488**	**MH453481**	**-**
** *Roussoellopsis macrospora* **	**MFLUCC 12-0005**	**-**	**KJ474847**	**KJ474855**	**KJ474862**
*Roussoellopsis* sp.	NBRC 106246	-	AB524626	**-**	**-**
*Roussoellopsis tosaensis*	KT 1659	-	AB524625	MG829199	AB539104
*Setoarthopyrenia chromolaenae*	**MFLUCC 17-1444**	**MT214344**	**MT214438**	**MT235768**	**MT235805**
** *Thyridaria acaciae* **	**CBS 138873**	**KP004469**	**KP004497**	**-**	**-**
** *Thyridaria broussonetiae* **	**CBS 141481**	**NR_147658**	**KX650568**	**KX650539**	**KX650586**
*Torula herbarum*	CBS 111855	KF443409	KF443386	KF443403	KF443396
** *Torula hollandica* **	**CBS 220.69**	**KF443406**	**KF443384**	**KF443401**	**KF443393**
** *Xenoroussoella triseptata* **	**MFLUCC 17-1438**	**MT214343**	**MT214437**	**MT235767**	**MT235804**

**Abbreviations: CBS:** Culture collection of the Westerdijk Fungal Biodiversity Institute, Utrecht, Netherlands; **CPC:** Culture collection of Pedro Crous, Netherlands; **IFRDCC:** Research Institute of Resource Insects, Chinese Academy of Forestry Culture Collection, Kunming, China; **GMB:** Herbarium of Guizhou Medical University, Guiyang, China; **GMBCC:** Guizhou Medical University Culture Collection, Guiyang, China; **KUMCC:** Kunming Institute of Botany Culture Collection, Kunming, China; **MAFF:** Ministry of Agriculture, Forestry and Fisheries, Japan; **MFLUCC:** Mae Fah Luang University Culture Collection, Chiang Rai, Thailand; **MUT:** Mycotheca Universitatis Taurinensis, Department of Life Sciences and Systems Biology, University of Turin, Turin, Italy; **NBRC:** Biological Resource Center, National Institute of Technology and Evaluation, Chiba, Japan; **UTHSC:** The University of Tennessee Health Science Center, Memphis, USA; **DDQ:** Dong-Qin Dai; **KT:** K. Tanaka.

**Table 4 jof-08-00532-t004:** Different parameters for ML analyses.

Analyses	Parameters	Value
ML	Final ML Optimization Likelihood	−30,685.326261
	No of characters	3372
	Alignment patterns	1513
	Proportion of Undetermined characters or gaps	29.97%
	Substitution model	GTR
	Tree length	3.629102
	Estimated base frequencies	A = 0.240311
		C = 0.267630
		G = 0.270582
		T = 0.221477
	Substitution rates	AC = 1.886299
		AG = 5.163488
		AT = 1.984078
		CG = 1.308792
		CT = 9.477974
		GT = 1.000000
	Gamma distribution shape parameter	α = 0.180205

**Table 5 jof-08-00532-t005:** Morphological comparison of *R. papillate* with *R. hysterioides* and *R. japanensis*.

Characters	*R. japanensis* [1]	*R. papillate* (in This Study)	*R. hysterioides* [39]
Ascostromata	500–2000 μm diam., immersed under a clypeus, raised, visible, black, dome-shape areas on host surface, uni-biloculate	250–350 μm high, 500–900 μm long, 500–700 μm wide deeply immersed under a brown area, becoming raised at maturity, ellipsoidal to irregular coriaceous, solitary to gregarious, brown, with black papilla, uniloculate	230–280 µm high, 2–2.5 mm wide, immersed, flattened at the base, multilocular
Locules	190–210 μm high, 500–560 μm diam., depressed globose with a flattened base, single or 2–3 grouped, ostiolate	200–300 μm high, 450–500 μm diam., solitary, subglobose, brown to dark brown, with a central ostiole	75–150 × 35–50 µm, with the ostiole erumpent through the host epidermis
Peridium (Wall of locules)	10–15 μm thick at sides, composed of 3–5 layers of polygonal flattened cells (3.5–12.5 × 1.5–2.5 μm), surrounded by wedge-shaped stromatic region (450–800 μm wide at sides) composed of rectangular to polygonal cells (3.5–15 × 4–10 μm)	9–20 μm wide, composed of 1–2 layers of *textura angularis*, thin-walled flattened at the base, light brown to brown	
Asci	107–132 × 8–9.5 μm, cylindrical, short pedicellate	108–125 × 7–10 μm, cylindrical, short pedicellate	105–120 × 4–6 µm, cylindrical, short- pedicellate
Ascospores	16–22 × 5.5–7 μm, uniseriate, fusiform to ellipsoidal, with a median septum, 2-celled, brown, rough-walled more or less, covered with longitudinal striations and surrounded by an entire sheath of 0.5–4 μm wide	15–17 × 5.5–7 μm, uniseriate, ellipsoidal to broad fusiform, 2-celled, constricted at the septum, brown to dark brown, with longitudinal striations, surrounded by a mucilaginous sheath	13–20 × 4–6 µm, uniseriate, overlapping, fusiform, uniseptate, constricted at the septum, brown, slightly pointed at the ends, upper cell larger, with striate ornamentation on surface

**Table 6 jof-08-00532-t006:** Base pair differences of ITS and RPB2 gene loci of *R. uniloculata* and other related species.

Species	ITS	*rpb2*
*R. chiangraina*	7.22% (33/457)	3.68% (34/925)
*R. kunmingensis*	5.29% (25/472)	4.61% (39/845)
*R. mediterranea*	4.26% (20/469)	3.84% (25/651)
*R. neopustulans*	5.56% (27/469)	3.80% (35/922)
*R. padinae*	4.71% (22/467)	3.78% (35/925)

**Table 7 jof-08-00532-t007:** Ascospores, host, and distribution comparison of eleven sequence lacking known *Roussoella taxa* with three new species in this study. “-”: not available in the protologue.

Taxa	Ascospores	Host	Known Distribution	References
*Roussoella aequatoriensis* Hyde	26–33 × 9–11 µm, fusiform-ellipsoidal, 1-septate, constricted at the septum, brown, with oblique wall, striations running the entire length of the ascospore and with yellow coloured mucilaginous, pad-like appendages at each end	Palm	Ecuador, Puerto Rico	[41]
*R. alveolata* Ju, Rogers, and Huhndorf	34–42 × 11–13 µm, with ridges between the longitudinal striations.	Bamboo	Indonesia (Java)	[41]
*R. angustispora* Zhou, Cai, and Hyde	24–28 × 6–8 µm, ellipsoid-fusiform, 1-septate, constricted at the septum, brown, with reticulate wall ornamentations	Bamboo (*Bambusa changii*)	China (Hong Kong)	[42]
*R. bambusae* (Pat.) Monod	23 × 5 µm, elliptical elongated, often acute at both ends, not constricted at the septum, colorless and surrounded by a fleeting hyaline sheath	-	-	[43]
*R. calamicola* Fröhl., Hyde, and Aptroot	20–27(–29.5) × 7–8.5 μm, ellipsoidal, 1-septatae, brown, verrucose, surrounded by a mucilaginous sheath	*Calamus*	Australian (Queensland)	[44]
*R. chilensis* (Speg.) Ju, Rogers, and Huhndorf	asci contain only four 20–25(–28) × 6–8 µm, ascospores with longitudinal wall striations This fungus is unique amongst Roussoella in having four ascospores per ascus	Bamboo (*Chusquea*)	Chile	[41]
*R. donacicola* (Speg.) Ju, Rogers, and Huhndorf	ascospores are (6–)6.5–8(–8.5) × 3–3.5 µm, with longitudinal striations	Bamboo (*Arundo*, *Phyllostachys*)	Argentina, France	[41]
*R. palmicola* Fröhl., Hyde, and Aptroot	12.5–24 × 2.5–4 μm, fusiform, 1-septate, brown, striate, with small pads of mucilage at both ends	Rattan (*Calamus flabellatus*)	Brunei	[44]
*R. saltuensis* Hyde	25–30 × 8–11 µm, overlapping ellipsoidal, 1-septate, constricted at the central septum, dark-brown, covered with irregular longitudinal striations and surrounded by a mucilaginous sheath which spreads in water	Palm (indet.)	Ecuador	[41]
*R. serrulata* (Ellis and Martin) Hyde and Aptroot	18–20 × 5–6 µm, characterized by the often deeply (up to 1 mm) immersed ascomata	Pam (*Serenoa serrulata*)	USA, Florida	[12]
*R. verruculosa* Cand. and Katum	7–8 × 5 µm, fusoid, septate, slightly constricted at the septum, rounded at the ends, verruculose	Bamboo (*Phyllostachys mitis*)	France	[45]
*R. papillate* Dai and Wijayaw	15–17 × 5.5–7 μm, brown to dark brown, rough-walled, with longitudinal striations	Bamboo	China (Yunnan)	In this study
*R. sinensis* Dai and Wijayaw	16.5–20.5 × 6–7.5 μm, ellipsoid to broad fusiform, upper cells bigger, constricted at the septum, narrowly at both ends, with longitudinal striations	Bamboo	China (Yunnan)	In this study
*R. uniloculata* Dai and Wijayaw	8.5–12× 3.5–4.5, ellipsoid to broad fusiform, 2-celled, upper cells bigger, occasionally curve, brown, constricted at the septum, with longitudinal striations	Bamboo	China (Yunnan)	In this study

## Data Availability

Not applicable.

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
