# Peer review of "Taxonomic and Phylogenetic Characterizations Reveal Four New Species, Two New Asexual Morph Reports, and Six New Country Records of Bambusicolous Roussoella from China"

_jof, 2022, doi:10.3390/jof8050532_

Round 1

Reviewer 1 Report

 minor corrections on the text

Author Response

Dear Reviewer,

Thank you a lot for reviewing our article. We corrected the MS based on the comments and prepared a separate response file, please see the attachment.

Best regards

Ms Han

Reviewer 2 Report

Good work with good efforts

Author Response

Dear Reviewer,

Thank you a lot for reviewing our article. We couldn't see any modified suggestions from you. Please feel free to contact us if you have any further comments that need to be revised.

Best regards

Ms Han   

Reviewer 3 Report

Title: Taxonomic and phylogenetic characterizations reveal four new species, two new asexual morph reports and six new country records of bambusicolous Roussoella from China

Authors: D.-Q. Dai, N. N. Wijayawardene, M. C. Dayarathne, J. Kumla, L.-S. Han, G.-Q. Zhang, X. Zhang, T.-T. Zhang, H.-H. Chen

Reference: jof-1738059

Article type: Research

Reviewer Comments:

The manuscript jof-1738059, entitled “Taxonomic and phylogenetic characterizations reveal four new species, two new asexual morph reports and six new country records of bambusicolous Roussoella from China”, describes the isolation and characterization of 24 specimens belonging to the family Roussoellaceae based on morphological and phylogenetic features. As a result, four novel lineages in Roussoella sensu stricto corresponding to four new species are presented.

General comments:

  1. Clarity and readability of the manuscript can be improved in terms of:
  2. English language
  3. Typographical errors
  4. Consistency
  5. The style used for reference citation and indication (e.g., lines 34-35: consider replacing “Liu et al. [1] introduced Roussoellaceae J.K. Liu et al. to accommodate three genera” by “The family Roussoellaceae accommodate three genera”)
  6. Simplify the name of the species by removing the name of the authors that described them (e.g., lines 35-36: please consider replacing “Neoroussoella J.K. Liu et al., Roussoella Sacc. and Roussoellopsis I. Hino & Katum” by “Neoroussoella, Roussoella, and Roussoellopsis”)
  7. Limit the number of references used to support each sentence (e.g., line 57: [1,5,15–17]). A maximum of three references should be used. Consider using Review articles.

Specific comments:

Line 21: please consider replacing “evidence” with “features”

Line 22: please consider replacing “LSU, rpb2 and tef1” with “LSU, rpb2, and tef1”

Lines 22-24: please consider rephrasing the sentence

Line 27: please consider replacing “culms of bamboo” with “bamboo culms”

Line 28: please consider replacing “Roussoella japanensis, R. nitidula, R. padinae, R. scabrispora and R. tuberculate” with “R. japanensis, R. nitidula, R. padinae, R. scabrispora, and R. tuberculate”

Line 36: please consider replacing “depend on” with “based on”

Line 37-39: please consider replacing “reinstated Roussoellaceae and treated that Roussoellaceae and Thyridariaceae are distinct families within Pleosporales, and this was proved in subsequent studies using additional taxa and combining morphological” with “reinstated Roussoellaceae; treating Roussoellaceae and Thyridariaceae as distinct families within Pleosporales. Subsequent studies confirmed this separation using additional taxa and combining morphological”

Line 47: please consider replacing “which was recorded from” with “collected from”

Lines 48-50: please consider replacing “Höhnel [11] found that Dothidea hysterioides Ces. is the older name for this taxon thus it was transferred to Roussoella as a new combination, i.e. Roussoella hysterioides (Ces.) Höhn.” with “Höhnel [11] proposed that Dothidea hysterioides Ces. is the former name for this taxon; thus, being transferred to the Roussoella genus, i.e., Roussoella hysterioides (Ces.) Höhn.”

Lines 50-51: please consider replacing “However, Aptroot [12], Müller and Arx [13] assigned Roussoella to Amphisphaeriaceae which comprises cylindrical” with “However, Aptroot [12] and Müller and Arx [13] assigned Roussoella to Amphisphaeriaceae, which comprises cylindrical”

Lines 52-54: please consider replacing “Aptroot [12] identified the asci of Roussoella as unitunicate and moved three species to this genus, while Aptroot [14] modified his concept of Roussoella and considered the asci to be bitunicate” with “Aptroot [12] described Roussoella asci as unitunicate and moved the three species to this genus, while Aptroot [14] modified his concept of Roussoella asci and considered them as bitunicate.”

Lines 54-57: please consider rephrasing the sentence

Lines 58-61: please consider replacing “Liu et al. [1] introduced the genus Neoroussoella as a monotypic genus and well distinguished from Roussoella in having uniloculate ascomata, absence of a clypeus and its asexual morph forming phoma-like conidia, with hyaline to pale brown, or brown, oblong to ellipsoidal, smooth-walled conidia [1,18].” with “Neoroussoella is a monotypic genus, distinct from Roussoella, with uniloculate ascomata, absence of a clypeus, and an asexual morph forming hyaline to brown, oblong to ellipsoidal, and smooth-walled phoma-like conidia [1, 18].

Line 61-63: Redundant sentence, please consider removing “Besides, Neoroussoella is characterized by a distinct asexual morph producing relatively smaller (3–4 × 1.5–2 μm), hyaline conidia with smooth walls [1].”

Lines 64-70: please consider replacing this paragraph with “Based on morphological characters and phylogenetic studies, the present study introduces four new species; R. multiloculate, R. papillate, R. sinensis, and R. uniloculata. The asexual morphs of R. kunmingensis and R. padinae isolated from dead bamboo culms (from a natural substrate) are also described for the first time. Finally, the species N. bambusae, R. japanensis, R. nitidula, R. padinae, R. scabrispora, and R. tuberculate are reported for the first time in China.”

Lines 73-86: please consider replacing this paragraph with “Bamboo culms were collected in Yunnan, China, and conserved in protection bags for two days until they arrived at the laboratory. Samples were examined, and single spore isolation was performed as previously described [17]. Morphological characters were examined using water slides and photographed (Olympus BX53 DIC compound microscope with an Olympus DP74 camera). Fruiting bodies were also photographed (Leica S8AP0 stereomicroscope with an HDMI 200C camera). Measurements were registered [Tarosoft (R) Image FrameWork 80 software]. Specimens and living cultures were deposited at the Herbarium of Guizhou Medical University (GMB) and Guizhou Medical University Culture Collection (GMBCC) in Guiyang, China. Duplicates of holotypes and ex-type cultures were also deposited at the herbarium of Research Institute of Resource Insects, Chinese Academy of Forestry (IFRD), and Research Institute of Resource Insects, Chinese Academy of Forestry Culture Collection (IFRDCC) in Kunming, China. Index Fungorum [20] numbers were provided for newly introduced taxa.”

Line 88-101: please consider replacing this paragraph with “Pure cultures were grown on PDA media, for 30–40 days, at 28ËšC, in the dark. Fresh mycelium was scraped using a surgical knife and placed into a 1.5 ml centrifuge tube, grinding into powder using liquid nitrogen. Genomic DNA was extracted following the instruction book of the Biospin Fungus Genomic DNA Extraction Kit (BioFlux® ). Information of primers used for the amplification of internal transcribed spacers (ITS), small subunit rDNA (SSU), large subunit rDNA (LSU), translation elongation factor 1-α gene region (tef1), and RNA polymerase II second largest subunit (rpb2) genes is presented in table 1. ITS, SSU, LSU, rpb2, and tef1 loci amplifications were performed by Polymerase Chain Reaction (Eppendorf Mastercycler nexus gradient) according to the conditions presented in Table 2 [17]. The PCR products were sequenced, and the sequences were deposited in GenBank, as shown in Table 3.”

Lines 107-135: please consider replacing the paragraphs by “The quality of the sequences company was verified (BioEdit v. 7.0 [27]), and alignments from single genes were generated (MAFFT v. 7.215 [26]) (http://mafft.cbrc.jp/alignment/server/index.html), being manually edited when needed (MEGA6 version 6.0 [28]). The combined alignment of multi-genes was carried out (MEGA6 [28]). Maximum-likelihood (ML) analyses were performed (software RAxMLGUI v.1.0. [29,30]) with a 1000 bootstrap. Multi-gene alignments were uploaded to the website (http://sing.ei.uvigo.es/ALTER/). The best nucleotide substitution model was determined using the online tool Findmodel (http://www.hiv.lanl.gov/content/sequence/findmodel/findmodel.html) and was executed in RAxMLGUI to generate the best ML.

Bayesian analyses were performed using MrBayes v. 3.0b4 [31]. MrModeltest v. 2.2 selected the best evolution model [32]. Posterior probabilities (PP) [33,34] were performed by Markov Chain Monte Carlo sampling (MCMC)[35]. Six simultaneous Markov chains were run for 1,000,000 generations, and trees were sampled every 100th generation. The burn-in was set to 0.25, and the run was automatically stopped when the average standard deviation of split frequencies reached below 0.01 [36]. Trees were constructed (TreeView [37]) and formatted (Adobe Illustrator CS v. 5). Maximum-likelihood bootstrap values (MLBP) equal to or greater than 50% and Bayesian posterior probabilities (BYPP) > 0.80 are given at the branches. The sequences used in this study are listed in Table 1. The alignment based on the combined loci and phylogenetic tree were submitted to TreeBASE under the code 29601 (http://purl.org/phylo/treebase/phylows/study/TB2: 29601).”

Lines 142-163: please consider replacing the paragraphs by “The sequence data set of combined ITS, LSU, rpb2, and tef1 loci were used to determine the phylogenetic position of the newly generated described strains. SSU sequences were not included in the alignment, as most Roussoella taxa lack SSU sequences in the GenBank. The dataset comprised 90 strains, including two outgroup strains (Torula herbarum CBS 220.69 and CBS 111855, Table 3). The final alignment comprises 3381 characters used for the phylogenetic analyses, including gaps. The RAxML analysis of the combined dataset generated a best-scoring tree with a final ML optimization likelihood value of 150 -30685.326261 (Table 4). GTR+I+G model was selected as the best model based on MrModeltest and was used for the Bayesian analysis. Based on the multi-gene phylogenetic analyses (Figure 1), 26 new isolates were grouped in the family Roussoellaceae (96% MLBP, 1.00 BYPP). Five isolates, GMBCC1056, GMBCC1065, GMBCC1069, GMBCC1071, and GMBCC1080, and one specimen, GMB1219, represented a novel species Roussoella multiloculate sp. nov. and formed a sister clade to R. verrucispora with high statistical support (100% MLBP, 1.00 BYPP). Roussoella papillate sp. nov. (GMBCC 1121 and IFRDCC 3103) grouped sister with R. japanensis and R. hysterioides with high bootstrap support (100% MLBP, 1.00 BYPP). The third new species, R. sinensis sp. nov., clustered together with R. doimaesalongensis, R. siamensis, and R. yunnanensis. The last new taxon, R. uniloculata, forms a distinct clade at the base of lineage, which contains R. angusta, R. chiangraina, R.kunmingensis, R. magnatum, R.mediterranea, R. neopustulans, and R. padinae. The R. sinensis and R. uniloculata clades are phylogenetically distant from the known species (Figure 1).”

No line number (discussion): please consider replacing the paragraphs by “The family Roussoellaceae (in Pleosporales), including the genera Neoroussoella, Roussoella, and Roussoellopsis [1], are saprobes in different hosts, especially bamboo and palms (terrestrial and aquatic environments) or human pathogens [40]. Recently, three new species have been isolated from marine environments [8]. Currently, the family comprises 12 genera reported as sexual, asexual, or holomorph [1,8,9,17,40,50]. This study introduced four new species of Roussoella: three reported as sexual morphs (i.e., R. papillate, R. sinensis, and R. uniloculata), and another as an asexual morph (i.e., R. multiloculate). All species have been reported as saprobic taxa on bamboo plants. 

Besides, asexual morphs of R. kunmingensis (described initially as a sexual taxon [8]) and R. padinae (described initially as without sexual or asexual characteristics fide [8]) were reported for the first time in the present study. Sexual and asexual links were established based on DNA sequence analyses (Figure 1). The sexual morph of Roussoella kunmingensis was also reported from Kunming, Yunnan Province. Nevertheless, Poli et al. [8] introduced R. padinae from brown alga Padina pavonica (Italy), thus from the marine environment. This finding indicates that Roussoella species present a broad range of habitats and distribution. Besides, the same species could be reported from different environments but as its alternative morph.

Neoroussoella bambusae (Thailand), R. japanensis (Japan), R. nitidula (Malaysia), R. scabrispora (Java), and R. tuberculata (Thailand) species were reported from China for the first time. These records confirmed that the members of Roussoellaceae have a broad range of geographical distribution in South East Asia and Central Asia. We predict that novel species could occur in other tropical Asian countries, such as India, Laos, Myanmar, Pakistan, and Sri Lanka. The genus Roussoella comprises 51 epithets [51], but only 46 species are listed in Species Fungorum [51]. Eight species were transferred to Dothideaceae, Thyridariaceae, Phyllachoraceae, and Diaporthales [2,7,25,41]. Besides, two species were transferred to Neoroussoella and Pseudoroussoella, which are also nested in Roussoellaceae [42]. Moreover, R. hysterioides var. minuta (I. Hino & Katum.) I. Hino & Katum. Listed as synonyms of R. hysterioides in Index Fungorum [20], two species R. phyllostachydis and R. minutella were synonymized as R. pustulans by Hyde [43]. Thus, total 33 taxa were accepted in Roussoellaceae. Hongsanan et al. [40] mentioned that DNA sequences data are available only for 22 species. Hence, it is essential to recollect known species lacking DNA sequence data and designate epitypes. In this study, we compared the morphological characters of the sequence lacking Roussoella taxa [12,43–48] prior to introducing the new taxa (Table 7).

Scientific comments: 

Line 97: Please elucidate if both Forward and Reverse primers were used for sequencing and if these primers were the same as those used for amplification.

Lines 108-112: this step is standard procedure. There is no need to be so exhaustive in the description.

Figures and Tables:

Table 3: Please consider removing the red color and use an asterisk (*). Also, consider moving this table to supplementary material.

Table 7: Please consider placing the table in the Results Section.

Author Response

(The authors gave the same response as above.)
